# Preserved cognitive functions with age are determined by domain-dependent shifts in network responsivity

Dávid Samu[1,2], Karen L. Campbell[2,3], Kamen A. Tsvetanov[1,2,4], Meredith A. Shafto[1,2], Cam-CAN[†] & Lorraine K. Tyler[1,2]

Healthy ageing has disparate effects on different cognitive domains. The neural basis of these differences, however, is largely unknown. We investigated this question by using Independent Components Analysis to obtain functional brain components from 98 healthy participants aged 23–87 years from the population-based Cam-CAN cohort. Participants performed two cognitive tasks that show age-related decrease (fluid intelligence and object naming) and a syntactic comprehension task that shows age-related preservation. We report that activation of task-positive neural components predicts inter-individual differences in performance in each task across the adult lifespan. Furthermore, only the two tasks that show performance declines with age show age-related decreases in task-positive activation of neural components and decreasing default mode (DM) suppression. Our results suggest that distributed, multi-component brain responsivity supports cognition across the adult lifespan, and the maintenance of this, along with maintained DM deactivation, characterizes successful ageing and may explain differential ageing trajectories across cognitive domains.

[1] Department of Psychology, University of Cambridge, Cambridge CB2 3EB, UK. [2] Cambridge Centre for Ageing and Neuroscience (Cam-CAN), University of Cambridge and MRC Cognition and Brain Sciences Unit, Cambridge CB2 3EB, UK. [3] Department of Psychology, Brock University, St. Catharines, Ontario L2S 3A1, Canada. [4] Department of Clinical Neurosciences, University of Cambridge, Cambridge, UK. Correspondence and requests for materials should be addressed to D.S. (email: dsamu@csl.psychol.cam.ac.uk).
[†]A full list of consortium members appears at the end of the paper.

A striking feature of normal ageing is its widespread but disparate effects on cognition, with several cognitive domains exhibiting decline (for example, memory and fluid cognition), while others are preserved (for example, language comprehension) or even improved (for example, crystallized intelligence, vocabulary) well into old age[1,2]. Concurrently, ageing is also accompanied by widespread changes in the brain, from diffuse grey matter atrophy[3] and loss of white matter (WM) integrity[4] to widespread changes in neurovascular coupling[5] and decreased segregation of large-scale functional networks[6]. Understanding the neurophysiological changes underpinning age-related cognitive impairment and preservation is crucial for devising neurobiologically informed interventions[7]. While the importance of this issue for our ageing societies is being increasingly recognized[8], our knowledge regarding normal age-related cognitive decline is still limited[7,9].

Much previous research on cognitive ageing has focused on a single task per study, which, in line with the concern of issue isolationism[10], has led to a number of neurocognitive theories of ageing[7]. Two dominant theories, functional compensation and maintenance, predict that radically different neural mechanisms are responsible for successful cognitive ageing. According to the brain maintenance hypothesis, successful ageing is underpinned by retaining youth-like neural structure and function[11,12]. The theory of functional compensation, on the other hand, posits the presence of functional reorganization in response to gradual loss in neural structure during the course of normal ageing[13,14]. While these earlier studies and the resulting pluralism of theories are important and continue to guide the field, here we advocate a multiple-task, multiple-domain approach in order to test whether such generic neural mechanisms underlie the pattern of similarities and differences in cognitive ageing seen across various domains[1,2]. Studies looking at preserved or improved cognition, and directly comparing domains with different ageing trajectories, would be particularly informative given the almost exclusive focus of previous research on declining cognitive abilities, such as fluid processing or working memory[15].

The activity of the human brain is globally organized in a set of large-scale networks[16–18], many of which have been successfully associated to specific sensory, motor, higher-order cognitive and control functions[19–21]. Despite the importance of these networks in brain function and cognition, however, our knowledge is still limited on their role in normal and successful cognitive ageing, especially across different cognitive domains. Previous studies have detected widespread age-related reorganization in these brain networks and their connectivity[16,22], and related such changes to cognition[23,24]. These studies, however, are either limited in the number and diversity of cognitive tasks they examined or lack the direct comparison between behaviour and online, task-based neuroimaging recordings. Again, we propose that multiple-domain studies on these brain networks, allowing for the comparison of their function across diverse tasks on the same set of individuals, are required to better characterize and refine the number of competing theories of neurocognitive ageing[7,12,13].

In the current study, according to the above proposal, we first test whether age differences in the activity of functional brain components could serve as the neural underpinnings of age-related cognitive differences, and then examine whether domain-specific variations in these age-related neural differences may explain the disparate effect of ageing on cognition across domains. To this end, we calculate task-evoked activity of functional brain components using a population-based ageing cohort covering the adult lifespan ($n = 98$, age: 23–87) from the Cam-CAN project[25], in each of three tasks: fluid intelligence and visual object naming (both typically exhibiting age-related cognitive decline) and syntactic comprehension (which tends to be preserved). We use spatial Independent Components Analysis (ICA)[26] to find the brain networks activated by task execution (task-positive networks[27]), either shared across or uniquely associated with each task (domain-general or domain-specific), as well as the prominent task-negative network, the default mode network (DM)[28–30].

In general, we hypothesized that age-related cognitive decline may result from functional disturbances in the task-positive components. In considering the complementary case, cognitive preservation, we test specific hypotheses of successful neurocognitive ageing, namely, whether high performance in older age relates to (1) some compensatory functional mechanism with ageing (functional compensation)[14], and/or (2) the ability to withstand normal age-related neural decline and maintain some (hypothetical) youth-like levels of neural function (brain maintenance)[12].

A related important question is whether the pattern of preserved versus declining cognition across tasks depends on the differential involvement of domain-specific components (for example, the frontotemporal language network[31]) versus domain-general components (for example, the multiple demand network[32]). Beyond the task-positive domain-general components, another pivotal domain-general network, the DM network[29,30,33], has been implicated in age-related declines across a variety of tasks[34–36], and may therefore play a central role in cognitive ageing in general. Specifically, we hypothesize that the extent and direction (activation or suppression) of DM responsivity demanded by the tasks may distinguish declining from preserved domains.

In general, by comparing age-related functional differences in task-positive and DM components across tasks, we hope to identify neural indicators differentiating declining and preserved cognitive domains, as well as the neurocognitive mechanisms underlying normal and successful cognitive ageing at the inter-individual level. To preface the main results, we find that (1) performance on Fluid Intelligence and Picture Naming shows an age-related decrease, while Sentence Comprehension performance is preserved; (2) in each task, inter-individual performance differences are best predicted by the task's task-positive components; (3) differences in the activity of these components also predict age-related cognitive differences both within and across tasks; (4) our results support the theory of functional maintenance, rather than compensation; and finally, (5) the declining tasks, but not the preserved task, are characterized by the suppression of the DM network, which suppression weakens in older participants. Altogether, our results suggest that successful cognitive ageing across multiple cognitive domains is supported by the maintenance of high neural responsiveness, and point to the age-related loss in the ability to modulate task-positive and task-negative DM activity as one of the primary domain-specific neural causes of age-related cognitive decline.

## Results

**Behavioural results.** In accordance with earlier studies, behavioural performance (task scores) exhibited strong age-related decrease for Fluid Intelligence[37,38] (Pearson's r [95% confidence interval (CI)]: $r(96) = -0.68$ [$-0.55$, $-0.77$], $P = 10^{-12}$) and Picture Naming[39,40] ($r(96) = -0.59$ [$-0.43$, $-0.70$], $P = 10^{-9}$), but no significant age-related difference for Sentence Comprehension[22] ($r(97) = -0.03$ [$-0.17$, $0.24$], $P = 0.74$) (Fig. 1, see also Supplementary Table 1 for detailed report on behavioural results). This difference in age effects across tasks forms the basis of our following analysis to find candidate domain-specific and domain-

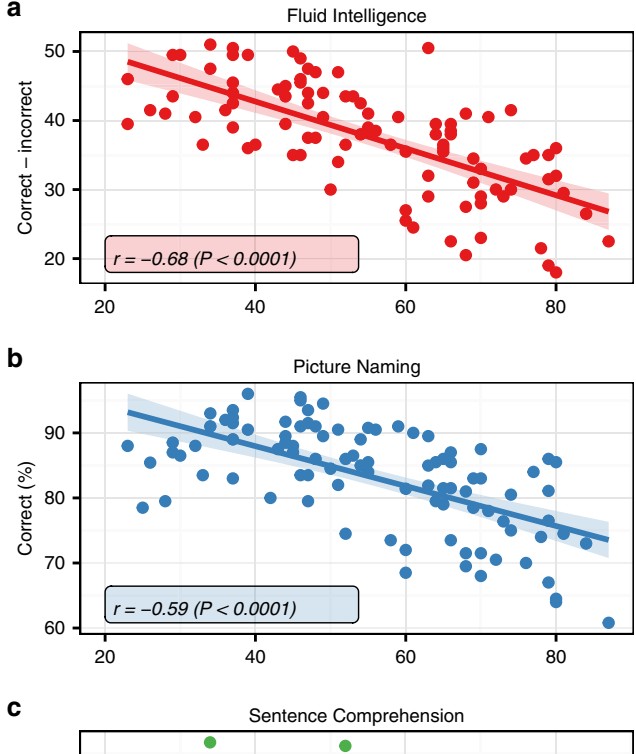

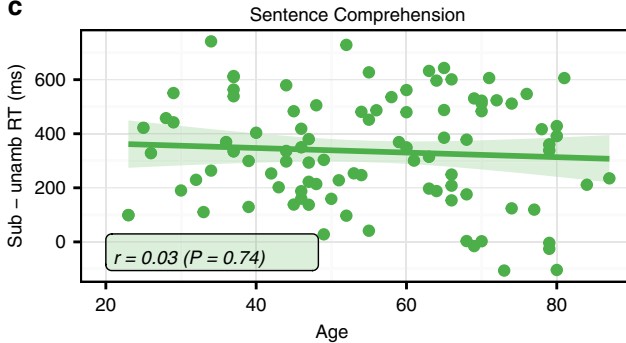

**Figure 1 | Behavioural scores.** Fluid Intelligence (**a**) and Picture Naming (**b**) show decline (negative age-related difference) in performance, while Sentence Comprehension score (**c**) is preserved across the lifespan.

general neural mechanisms that support declining versus preserved cognition with age.

**Brain components and their functional responsivity.** We found the optimal joint ICA decomposition of the tasks at $n = 50$ components (see Methods). Excluding noise-related, vascular and primary motor components (see Methods) resulted in 30 neural components (see Supplementary Table 2). After fitting the task events to the task-specific time-course of each component, we calculated a functional responsivity index for each component in each task (see Methods). Given the conditions of interest used for the calculation (more demanding and less demanding conditions), these responsivity values represent the components' excess activation/suppression evoked by the central cognitive components of interest in each task (rather than being the neural correspondences of the tasks' sensory-motor elements peripheral to our interest here).

Across-subject mean loading values of each component are shown in Supplementary Figs 3–7. In general, most components showed significant, either positive or negative, mean responsivity to task demand in each task, confirming our experimental manipulations. Responsivity to Fluid Intelligence was significantly higher than to the other two tasks (mean absolute responsivity

values across all subjects and components [95% CI]: Fluid Intelligence: 0.226 [0.171, 0.281], Picture Naming: 0.058 [0.039, 0.076], Sentence Comprehension: 0.037 [0.027, 0.046]), likely due to the more powerful block design used for the Fluid Intelligence task (see Methods), as opposed to the event-related design used for Picture Naming and Sentence Comprehension[41].

We also found strong age-related differences in the responsivity values of many components (Fig. 2a–c), especially in the declining tasks. In particular, the two declining tasks, in contrast to the preserved task, showed both stronger age-effects on individual component responsivity (Fig. 2a–c, $y$ axis), as well as a significantly stronger tendency for more responsive (activated or suppressed) components to show greater age-related decrease than less responsive ones (compare trend lines on Fig. 2a,b with Fig. 2c, difference between correlations [and 95% CI in $r$-value difference]: Fluid Intelligence–Picture Naming: $z(95) = 0.68$ [−0.02, −0.03], $P = 0.50$; Fluid Intelligence–Sentence Comprehension: $z(95) = 3.67$ [0.21, 0.22], $P = 2 \times 10^{-4}$; Picture Naming–Sentence Comprehension: $z(96) = 4.36$ [0.24, 0.25], $P = 7 \times 10^{-6}$). Notably, these results are in accordance with the age-related behavioural differences in the tasks (see Fig. 1).

**Brain responsivity predicts performance across the lifespan.** We first identified, for each task, the set of components that contributed the most to its execution, the task-positive components[27]. To do this we used the heuristic that more responsive components are more related to task performance (see Methods). In this section, we first test the validity of this heuristic in our data, and then use it to identify a set of task-positive components which contribute the most to performance in each task. These components will then be used to produce a summary measure of task-specific neural function, mean task-positive responsivity (MTR, see Methods).

We first tested the heuristic that mean responsivity is a proxy for the component's contribution to task execution, that is, that more responsive components are more related to task performance than less responsive ones (see Methods). For each task, we observed a close relationship between the components' responsiveness and relation to performance (Fig. 2d–f). Furthermore, we found this generic relationship to hold also within each age-group of younger, middle-aged and older participants, and also in the entire cohort when controlling for age (Fig. 2g–i), suggesting that mean component responsivity is a good proxy for the component's involvement in task execution and performance, across cognitive domains and the entire adult lifespan.

Having seen its validity in the current data set, we next used the heuristic in order to find each task's task-positive components, that is, the set of components that collectively contribute the most to the execution of the task. When iteratively calculating the mean responsivity of a growing set of the most responsive components (multiple component responsivity, see Methods), we found that the average responsivity of $n = 4$ of the most responsive components is the most predictive of task performance in each task. This indicates that the four most responsive components in each task have the highest collective contribution to successful task execution, rendering them a good candidate set of task-positive components. Moreover, including age as a covariate into this analysis did not change the set of components implicated in each task, suggesting that the identified set of task-positive components are fairly robust across the adult lifetime (Fig. 3). Relatedly, when focusing on the mean responsivity of the four task-positive components (MTR) of each task, we found that the relation between MTR and task score was not moderated by age in any of the tasks (partial $R$-squared of MTR × score interaction term in a multiple linear regression model predicting task score: Fluid Intelligence: $R^2(96) = 0.5\%$, $P = 0.50$; Picture

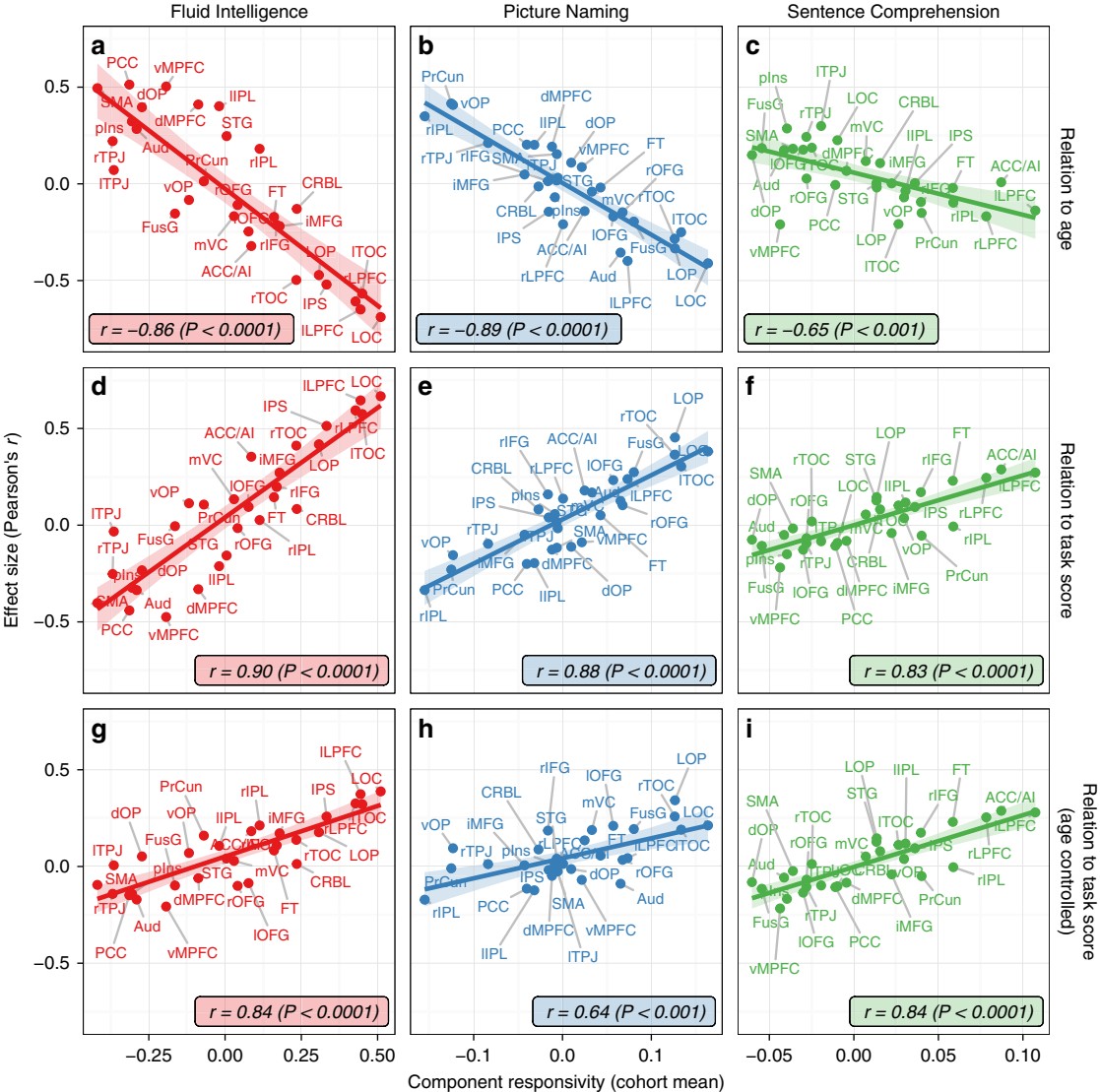

**Figure 2 | Relation of component responsivity to age and performance.** Cohort-mean component responsivities (x axis) are correlated against three effects on subject-specific responsivity values (y axis, see labels on the right side): relation (correlation) to age (**a–c**), task score (**d–f**) and task score when controlling for age (**g–i**). (**a–c**) More responsive components (to the right along x axis) show more age-related decrease in responsivity (more negative values along y axis) especially in the declining tasks. (**d–i**) More responsive components are more related to task score in all tasks (second row), largely independently of age-related difference in both responsivity and behaviour (third row). Component labels are given in each panel (see Supplementary Table 2).

Naming: $R^2(96) = 0.0\%$, $P = 0.90$; Sentence Comparison: $R^2(97) = 2.8\%$, $P = 0.11$), suggesting a stable association between task-positive responsivity and performance across the adult lifespan.

The obtained sets of task-positive components are in line with established findings (Figs 4, 5a–c). Task-positive components of Fluid Intelligence comprise two higher-order visual components, lateral occipital and left temporo-occipital cortices (LOC and lTOC) and key regions of the multiple demand network, left and right lateral prefrontal cortices (lLPFC and rLPFC), responsible for flexible executive control and fluid processing[32]. The four vision-related components of Picture Naming, LOC and lTOC, shared with Fluid Intelligence, and the lateral occipital pole and the right temporo-occipital cortex (LOP and rTOC), altogether cover much of the ventral visual stream involved in object representation and recognition[42]. Finally, Sentence Comprehension shares lLPFC and rLPFC with Fluid Intelligence, along with a dorsal anterior cingulate and anterior insula

component (ACC/AI), known as the salience network[43], and a fourth component covering the left inferior frontal gyrus (BA44 and 45) and anterior regions of the left superior and middle temporal gyri, known as the frontotemporal (FT) network of language comprehension[31,44].

Having identified the task-positive components of each task and established their relation to cognition across the lifespan, we further characterized the association among age, task-positive responsivity and cognition across tasks.

**Brain responsivity differentiates domains by age-effect.** As discussed in the section Brain components and their functional responsivity, we found that more responsive components not only contribute more to task performance (Fig. 2d–i), but, in the declining tasks, also demonstrate a stronger age-related decrease in responsivity (Fig. 2a,b). In particular, for the crucial task-positive components, MTR of the cognitively declining tasks also declines strongly with age (Pearson's r [95% CI]: Fluid

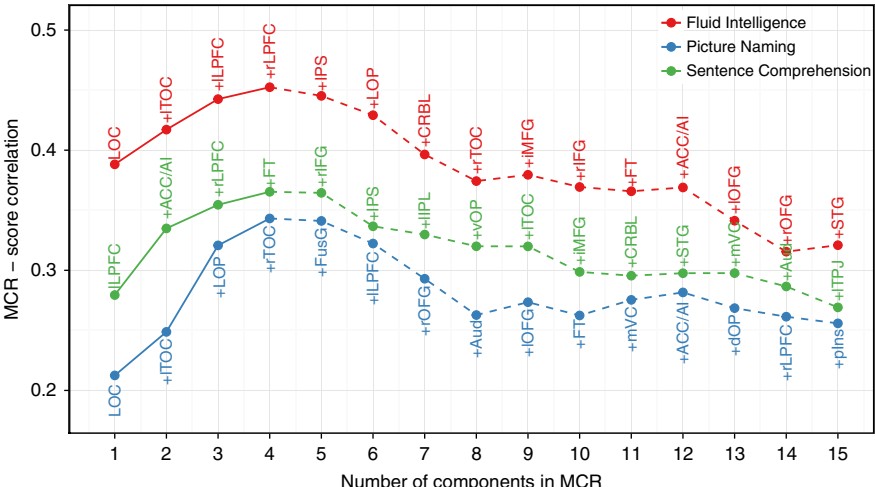

**Figure 3 | Task performance and multiple component responsivity across varying number of components.** For accessibility, correlation between multiple component responsivity (MCR) and task score is shown only up to the $n = 15$ most responsive components in figure (after which correlation continues to decline until the end at $n = 30$ components). Component, from left to right, are in the order of their addition to MCR, denoted by the '$+$' signs (for example, the MCR of Fluid Intelligence at $n = 4$ is the mean responsivity across LOC, ITOC, lLPFC and rLPFC). In each task, MCR optimally predicts task performance at $n = 4$ of the most responsive components. In order to isolate the effect of age and find a canonical (age-independent) set of task-positive components, the figure shows results after controlling for age. The same test without controlling for age, however, yields qualitatively the same results.

Intelligence: $r(96) = -0.72$ $[-0.61, -0.81]$, $P = 10^{-16}$; Picture Naming: $r(96) = -0.42$ $[-0.24, -0.57]$, $P = 2 \times 10^{-5}$; Fig. 5d,e), suggesting that MTR may act as a neural mediator mechanism of age-related cognitive decline. Our mediation analysis (see Methods) provided further evidence for this possibility: the MTRs of the declining tasks were found to account for a large portion of the shared variance between age and cognition (proportion of shared variance explained by MTR [95% CI]: Fluid Intelligence: 48% [25%, 76%], $P < 0.01$, $n = 96$; Picture Naming: 21% [7%, 43%], $n = 96$, $P = 0.01$). While keeping the limitations of mediation models of cross-sectional cohorts in mind (see Methods and also ref. 45), these results collectively point to brain responsivity as a potential neural mediator of some of the effect of age on cognition.

In contrast, MTR did not show age-related differences in the cognitively preserved Sentence Comprehension task (Pearson's $r$ [95% CI]: $r(97) = -0.11$ $[-0.30, 0.10]$, $P = 0.30$, Fig. 5f). We note here that the association between MTR and task score, when assessed across the lifespan without controlling for age, is stronger in the declining tasks than in the preserved task (Fig. 5g–i). This difference across tasks, however, is attributable to the simultaneous age-related decrease in MTR and task score in the declining tasks (Fig. 5d–f), reinforcing our finding about the age-independent relation between neural responsivity and performance present in all tasks to a similar same degree (see Fig. 5j–l).

Together, these findings suggest that the cognitively preserved nature of language comprehension is supported by the domain-specific maintenance of functional responsivity, while the strong age-related decrease (lack of maintenance) in responsivity in the other two tasks may be responsible for their cognitively declining nature.

Having established the general relationship between age, neural responsivity and cognition, we next tested specific hypotheses of successful cognitive ageing, namely functional compensation and maintenance.

**Functional compensation.** The notion of (age-related) functional compensation is typically described as a positive association

between increased neural activity and cognitive performance in older adults[13,14]. We tested the presence of potential functional compensation mechanisms by assessing two indicators of this complex effect.

First, we tested the presence of any age-related increase in positive component responsivity. Such an effect would indicate excess neural recruitment in older adults, the relevance of which to cognition would need to be tested subsequently (see next paragraph). In this test, correlating component responsivity with age in each task, we found no evidence for this potential positive recruitment effect in any of the positively activated components in any of the tasks ($P > 0.01$ for all tests, without multiple comparisons correction, older-age group with $n = 34$, for detailed results see Supplementary Table 3, Test 1).

Then, we tested for potential compensatory mechanisms in the form of an increasing association between component responsivity and cognition with age. We note that, although such effect is conceivable even in the absence of an age-related increase in component responsivity (see previous paragraph), typically the presence of both effects are required in the standard notion of compensation[13]. Formally, we tested, for each component and task, whether the relation between component responsivity and performance is moderated positively by age (see Methods). We found no significant moderation effect in any of the components and tasks ($P > 0.01$ for all tests, without multiple comparisons correction for $3 \times 30$ tests, older-age group with $n = 34$, for detailed results see Supplementary Table 3, Test 2). In sum, we found no evidence for either any increased functional recruitment or performance-related functional compensation mechanism in the current set of tasks and components.

**Functional maintenance.** Next, we tested an alternative hypothesis of successful cognitive ageing, functional maintenance, which claims that older adults possessing more youth-like neural characteristics are better able to withstand cognitive decline[11,12]. To this end, we assessed whether MTR, which shows strong age-related decrease in the declining tasks (see Fig. 5d,e), can also explain inter-individual cognitive differences in older age. Similarly to the results on the entire cohort (see section Brain responsivity predicts performance across the lifespan),

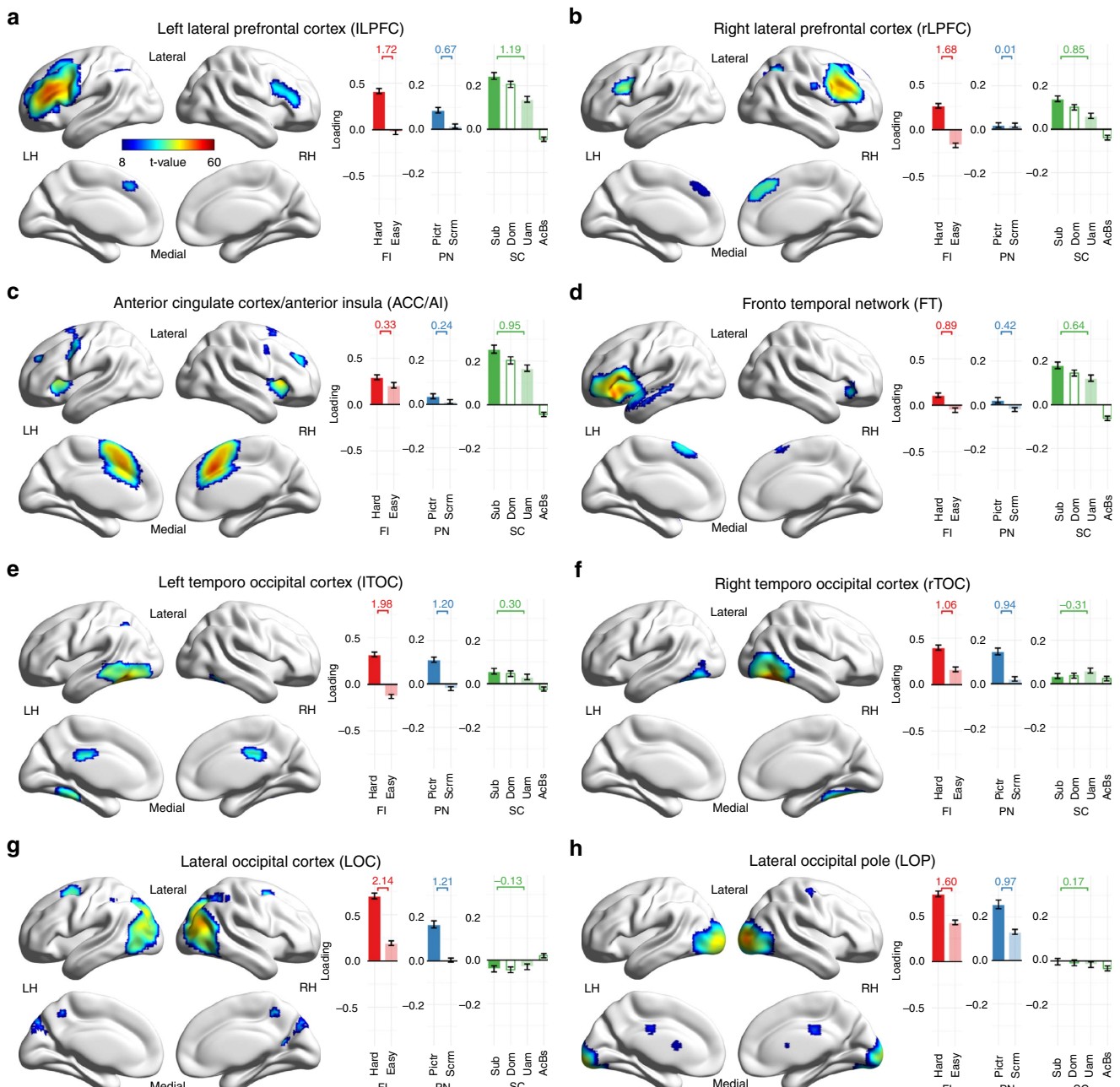

**Figure 4 | Spatial maps and loading values of task-positive components.** (**a**–**h**) All task positive components combined across the three tasks. Task-positive components of Fluid Intelligence: lLPFC, rLPFC, lTOC and LOC (**a**,**b**,**g**); Picture Naming: lTOC, rTOC, LOC and LOP (**e**–**h**); Syntactic Comprehension: lLPFC, rLPFC, ACC/AI and FT (**a**–**d**). Spatial maps (on the left) are group-level thresholded t-maps. Bars (and whiskers) on the right denote cohort-mean loading values (± s.e.m.), sorted by task (FI: Fluid Intelligence, PN: Picture Naming, SC: Sentence Comprehension), and higher (dark colour) versus lower (light colour) cognitive load (Hard: hard puzzle, Easy: easy puzzle, Pictr: picture naming, Scrm: scrambled image, Sub: subordinate sentences, Dom: dominant sentences, Uam: unambiguous sentences, AcBs: acoustic baseline). Across-condition effect sizes (mean over SD of condition contrast) are given above the bars. LH/RH: left/right hemisphere.

MTR correlated significantly to task score when tested in the older-age group only (Pearson's $r$ [95% CI]: Fluid Intelligence: $r(34) = 0.37$ [0.01, 0.65], $P = 0.043$; Picture Naming: $r(34) = 0.39$ [0.03, 0.66], $P = 0.032$). Altogether, these results indicate that high performing older adults also tend to possess more youth-like functional characteristics (higher task-positive responsivity) in the declining tasks, providing evidence for the functional maintenance hypothesis.

To find more specific functional characteristics differentiating declining versus preserved tasks, we next investigated potential functional differences among shared task-positive components, the putative domain-general components, across tasks.

**Domain-specific resilience of domain-general components.** A possible distinguishing factor between the preserved and declining tasks could be the domain-specific functional preservation of shared task-positive components, the domain-general components. Two frontal lobe components, left and right LPFCs, were shared task-positive components of the Fluid Intelligence and Sentence Comprehension tasks (see Figs 4 and 5a,c). Left LPFC also showed moderate correlation to Picture Naming

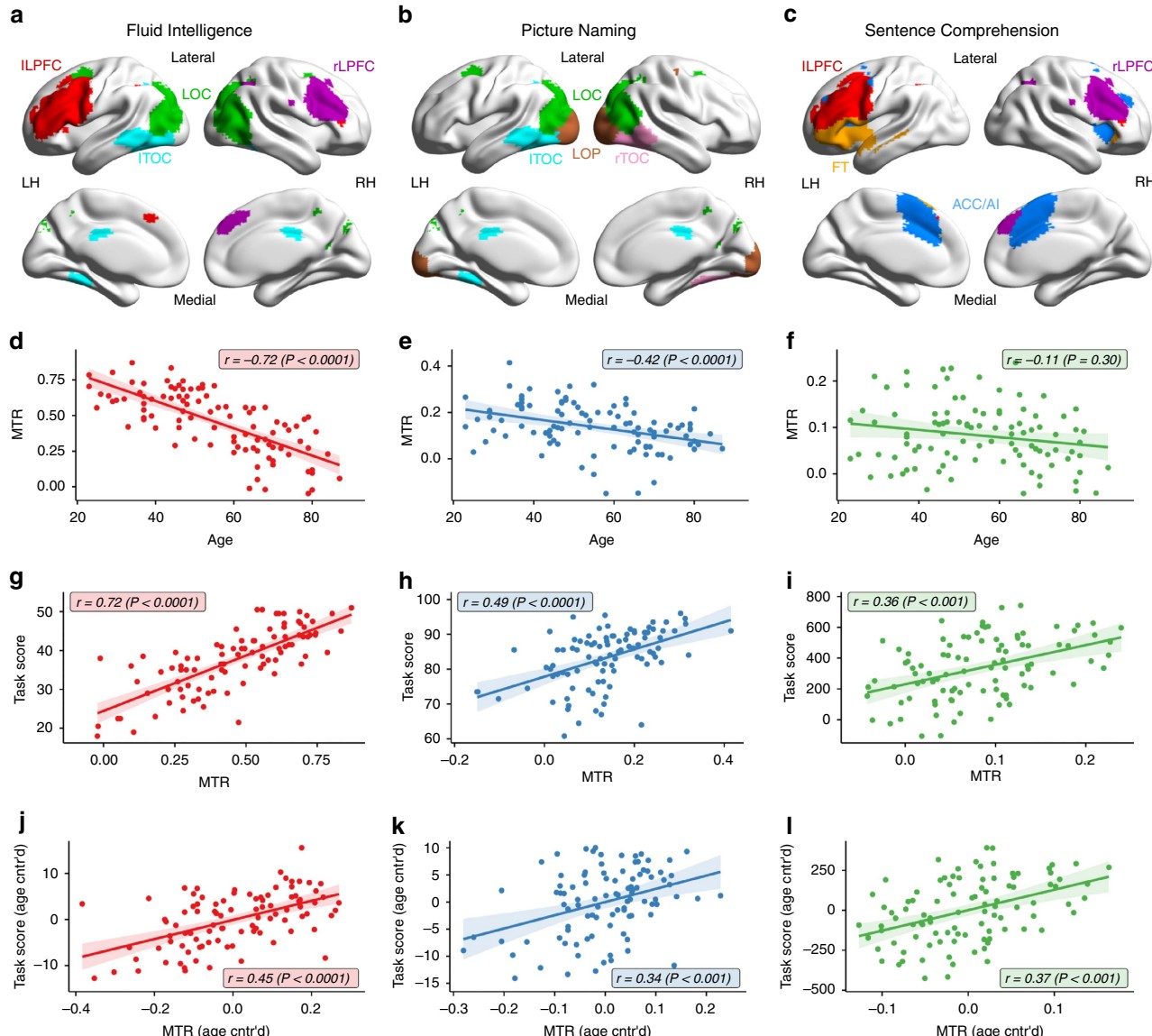

**Figure 5 | Relation of task-positive responsivity to age and performance.** Mean task-positive responsivity (MTR) predicts task performance across tasks, but shows age-related decreases only for the cognitively declining tasks. Results are grouped column-wise by task (see task labels on top). (**a–c**) Spatial maps of task-positive components of each task. Voxels are colour coded by corresponding component, see colours of inserted component labels. (**d–l**) Subject-level results per task. Dots correspond to results of individual participants, colour-coded by task (see labels on top). (**d–f**) Age-related difference of MTR. (**g–i**) Correlation between MTR and task score. (**j–l**) Correlation between MTR and task score, when controlling for age. LH/RH: left/right hemisphere.

performance (Pearson's $r$ [95% CI]: $r(96) = 0.24$ [0.04, 0.42], $P = 0.021$), though neither of these components was included to the task-positive components for Picture Naming (see Figs 3 and 5b).

To look for domain-specific functional resilience effects, we tested whether the age-effect on the responsivity of these shared LPFC components differs across task. We found strong age-related decrease in the responsivities of both LPFCs in Fluid Intelligence (Pearson's $r$ [95% CI]: lLPFC: $r(96) = -0.65$ [$-0.52$, $-0.75$], $P = 10^{-12}$; rLPFC: $r(96) = -0.61$ [$-0.46$, $-0.72$], $P = 10^{-10}$), and even in the only partially activated left LPFC in Picture Naming ($r(96) = -0.40$ [$-0.21$, $-0.56$], $P = 6 \times 10^{-5}$), while LPFC responsivities were largely preserved in Sentence Comprehension (lLPFC: $r(97) = -0.14$ [$-0.33$, 0.07], $P = 0.18$; rLPFC: $r(97) = -0.17$ [$-0.36$, 0.03], $P = 0.10$). These differences in correlations were significant both between Fluid Intelligence and Sentence Comprehension

(lLPFC: $t(95) = -5.21$, $P = 5 \times 10^{-7}$; rLPFC: $t(95) = -4.11$, $P = 4 \times 10^{-5}$) and between Picture Naming and Sentence Comprehension (lLPFC: $t(95) = -1.98$, $P = 0.026$), indicating that the difference in the declining or preserved nature of the tasks may be associated with the context-specific functional maintenance, or lack thereof, of the same, domain-general LPFC components.

Finally, to further explore these concurrent domain-specific functional and cognitive resilience effects, we next asked whether they are related to the activity of another prominent context-sensitive brain network, the DM network.

**DM suppression decreases with age in declining tasks.** To examine task-specific differences in DM activity, we first identified six components (Figs 6 and 7a) with high spatial overlap with canonical DM regions (Supplementary Table 2): left and right intra-parietal lobe (lIPL and rIPL), ventral and dorsal medial

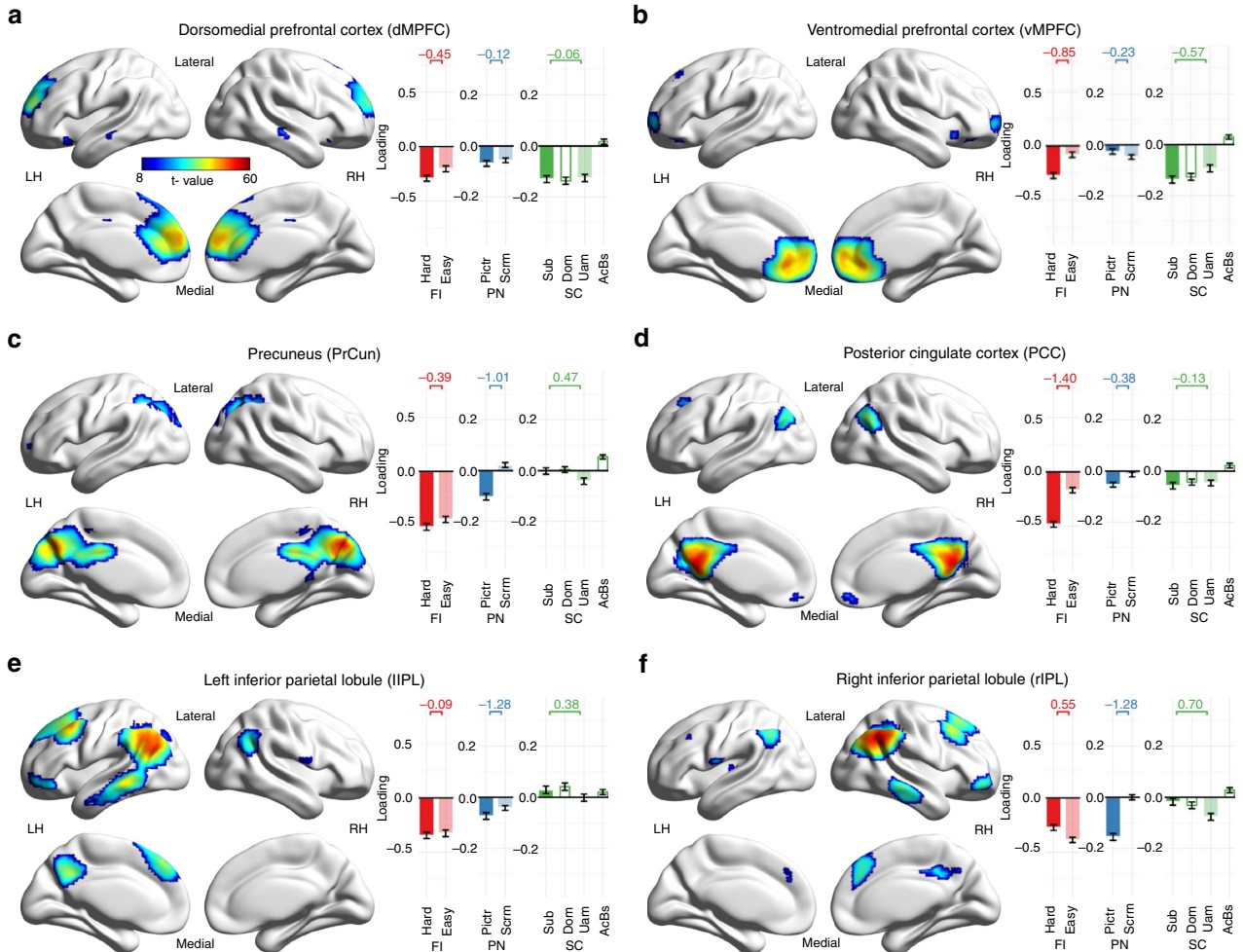

**Figure 6 | Spatial maps and loading values of default mode components. (a–f)** The six identified default mode components. Spatial maps (on the left) are group-level thresholded t-maps. Bars (and whiskers) on the right denote cohort-mean loading values (± s.e.m.), sorted by task and higher (dark colour) versus lower (light colour) cognitive load (see Fig. 4 for abbreviations). Across-condition effect sizes (mean over s.d. of condition contrast) are given above the bars. LH/RH: left/right hemisphere.

prefrontal cortex (vMPFC and dMPFC), posterior cingulate cortex (PCC) and the precuneus (PrCun). When calculating the mean responsivity scores of these DM components, we observed qualitative differences across tasks: while DM components were, on average, significantly suppressed in Fluid Intelligence (mean responsivity [95% CI]: $-0.095$ [$-0.123$, $-0.067$], one-sample two-sided $t$-test: $P = 10^{-9}$, $n = 96$) and Picture Naming ($-0.057$ [$-0.071$, $-0.043$], $P = 10^{-12}$, $n = 96$), they exhibited weak activation in Sentence Comprehension (0.012 [0.002, 0.022], $P = 0.023$, $n = 97$).

Next, we tested whether these differences in DM activity (strong suppression or weak activation) were complemented by differences in the functional connectivity (correlation between each component-pair's activity, see Methods) between task-positive and DM components. When calculating the mean functional connectivity between the two groups of components (FC, average over all subjects and every connections inter-connecting task-positive and DM component-pairs), we found significant functional segregation (negative FC) between task-positive and DM components in the declining tasks (FC [95% CI], Fluid Intelligence: $-0.062$ [$-0.080$, $-0.043$], one-sample two-sided $t$-test: $P = 10^{-8}$; Picture Naming: $-0.048$ [$-0.062$, $-0.033$], $P = 10^{-9}$, both $n = 96$), but the opposite, significant functional integration (positive FC) in the preserved Sentence Comprehension task (0.042 [0.026, 0.058], $P = 10^{-6}$,

$n = 97$) (Fig. 7e–g). These results indicate an opposing functional role of the task-positive and DM components in the declining tasks, and the lack of such functional antagonism in the preserved Sentence Comprehension task.

When testing for age-related differences in DM activity, we found a significant age-related decrease in the extent of DM suppression (that is, less negative responsivity with age) in Fluid Intelligence (Pearson's $r$ [95% CI]: $r(96) = 0.53$ [0.36, 0.66], $P = 10^{-8}$) and Picture Naming ($r(96) = 0.40$ [0.22, 0.56], $P = 6 \times 10^{-5}$), but no effect of age on DM responsivity in Sentence Comprehension ($r(97) = -0.12$ [$-0.31$, 0.09], $P = 0.26$) (Fig. 7b–d). Additionally, DM suppression correlated to task-positive activation (MTR) both in Fluid Intelligence (Pearson's $r$ [95% CI]: $r(96) = -0.45$ [$-0.60$, $-0.27$], $P = 5 \times 10^{-6}$), and Picture Naming ($r(96) = -0.33$ [$-0.50$, $-0.14$], $P = 0.001$), but not in Sentence Comprehension ($r(97) = 0.08$ [$-0.12$, 0.28], $P = 0.42$). As a consequence, similarly to MTR, DM suppression also correlated to task performance in the declining tasks (Pearson's $r$ [95% CI]: Fluid Intelligence: $r(96) = -0.35$ [$-0.52$, $-0.16$], $P = 6 \times 10^{-4}$; Picture Naming: $r(96) = -0.33$ [$-0.50$, $-0.13$], $P = 0.001$), but not in the preserved task (Sentence Comprehension: $r(97) = -0.09$ [$-0.29$, 0.11], $P = 0.37$). Unlike MTR, however, DM suppression did not correlate significantly to performance over and above age for any of the three tasks, and therefore was not

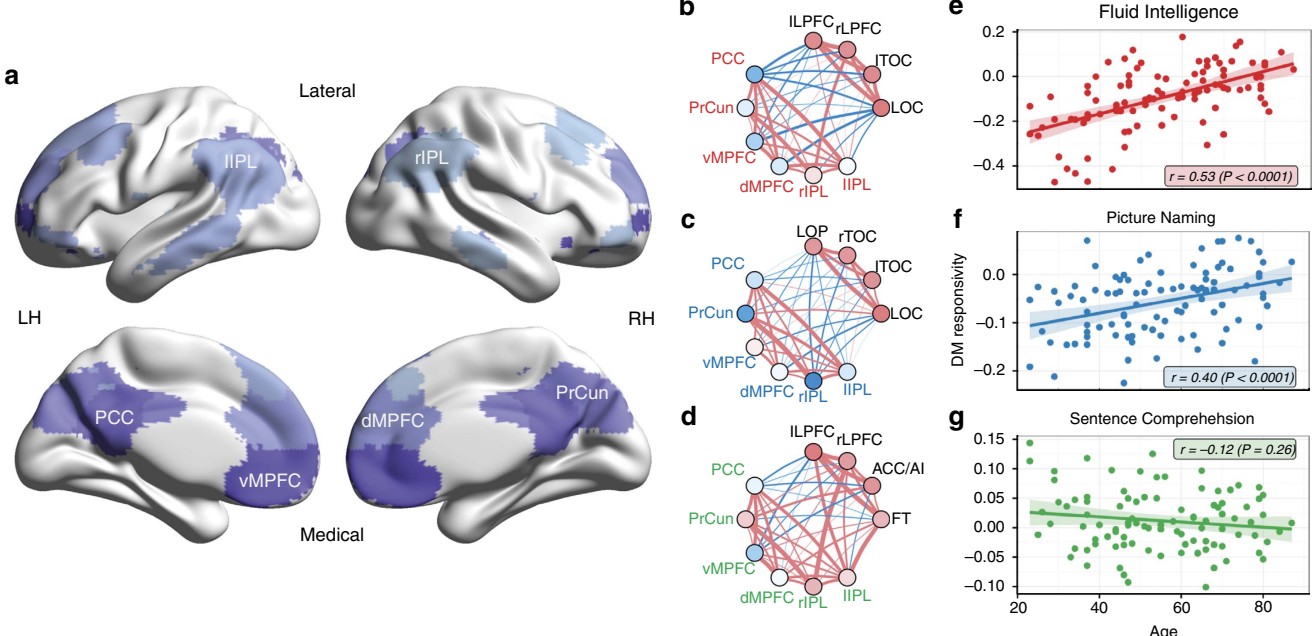

**Figure 7 | Responsivity and functional connectivity of default mode components.** Declining tasks involve default mode (DM) suppression, which shows age-related reduction. (**a**) Spatial maps of DM components. Colour shades represent different components (see inserted component labels). (**b–d**) Mean responsivity and functional connectivity (FC) of task-positive and DM components for each task. Disks represent components, coloured by their mean responsivity in the corresponding task (red: positive, blue: negative). Links represent FC between component pairs, coloured by sign of connectivity (red: positive, blue: negative), and with width proportional to the magnitude (absolute value) of correlation. In every task, both task-positive and DM components are strongly and positively connected among themselves (within group FC), whereas their interconnectivity (between groups FC) is dominantly negative for the declining tasks, but highly mixed for Sentence Comprehension. (**e–g**) Mean DM responsivity in each task as a function of age.

appropriate to be tested as a potential mediator between age and cognition.

This difference between the declining and the preserved tasks (lack of any significant DM suppression and functional segregation from task-positive components in Sentence Comprehension) may be explained by the partial involvement of some DM components in Sentence Comprehension. Specifically, while only vMPFC showed significant suppression in Sentence Comprehension, dMPFC and PCC were not modulated significantly by the task, and three other DM components, lIPL, rIPL and PrCun, exhibited moderate, but significant activation (or decrease in suppression) during the task (Fig. 6). In contrast, only one DM component showed (a small but significant) activation in each of the declining tasks, with four DM components exhibiting strong suppression. To control for the potential involvement of some of the six original DM components in Sentence Comprehension, we repeated all DM analysis using only the four medial DM components, but excluding the two lateral components (left and right IPLs), which had the highest responsivity during Sentence Comprehension. This set of analyses resulted in qualitatively the same results as those using all six DM components (see previous paragraph), supporting the robustness of our findings against the choice of DM components.

Altogether, these findings point to domain-specific differences in DM activity and functional connectivity, which, via the observed age-related decrease in DM suppression, may contribute to the observed discrepancies in the age-related cognitive differences across domains.

## Discussion

Successful execution of complex cognitive tasks requires the temporally orchestrated co-activation of a specific set of functionally segregated brain regions[46]. Previous studies indicated age-related differences in the activation of these functional components in a number of tasks[35,36,47], leading to the hypothesis that age-related cognitive changes may depend on functional disruptions in brain regions critical to the task. In this study, we investigated this general hypothesis across three tasks, two of which typically shows age-related cognitive decline and the third typically being preserved, in a large, population-based cohort covering the entire adult lifespan.

After obtaining functional brain components across the three tasks, we found that, in each task, a specific set of task-positive components support cognitive performance across the adult lifespan. Importantly, age-related cognitive decrease in both declining tasks was significantly accounted for by age-related decrease in the responsiveness of these key components. Moreover, the declining tasks were associated with strong suppression of the DM components, and this suppression also showed an age-related decrease. In contrast, in our third task, Sentence Comprehension, both performance and task-positive functional responsiveness were preserved with age, and, similarly, DM components showed neither a baseline suppression effect nor any age-related difference in responsivity. Collectively, these results point to a potentially general neural mechanism—the ability to recruit task-specific networks and suppress task-irrelevant DM regions in certain contexts—that does not only facilitate successful cognition across the adult lifespan, but may also explain differences in age-related cognitive decline or preservation across domains.

Functional compensation or reorganization has been suggested as a counterbalancing neural mechanism against the gradual loss of brain mass and integrity during the course of normal ageing[13,14]. In this study, we found no evidence for such compensation in the investigated tasks. On the other hand, our

results supported the alternative functional maintenance hypothesis[11,12] by pointing to the importance for older adults to retain the highly dynamic responsivity of both task-positive and DM components generally found in younger adults.

The multiple-task approach we advocate further allowed us to compare the responsivity of the same set of components across different domains, thereby identifying the left and right LPFCs as putative domain-general components shared across the tasks. These domain-general components exhibited domain-specific functional preservation, with their responsivity showing age-related decrease in the declining tasks but no age-related difference in the preserved task. Altogether, these findings indicate that domain-specific preservation of cognition may be underlain by some context-sensitive neural processes involving functionally central, domain-general brain components.

A key component of such a process could be another pivotal domain-general network, the DM network, the role of which is increasingly being recognized in both cognition and ageing[28–30]. Indeed, we found qualitative differences between the declining and preserved tasks in both the associated DM activity (strong suppression or weak activation) and the age-related differences it exhibited (age-related weakening of suppression or no age-related change). While some of these results may be explained by the partial involvement of some canonical DM components in the preserved language comprehension task[22,31,44], altogether, these findings suggest that older adults may find it particularly challenging to selectively suspend DM activity, while recruiting task-positive components, potentially leading to a (domain-specific) decline in cognition.

Because DM suppression is associated with tasks that have high externally focused attention requirements[28–30], our results suggest that these tasks are especially susceptible to age-related cognitive decline as the age-related reduction in DM suppression may interfere with the tasks' high external attention requirements. Similar age-related reduction in DM suppression has been reported previously for a wide range of declining tasks, including semantic word classification[36], visual and semantic memory task[35], and perceptual matching and attentional cueing[34]. Similar, but generally stronger effects were observed in Alzheimer's disease[48], pointing to the potentially widespread and pervasive negative consequences of ineffective DM suppression on cognition with ageing. Therefore, we propose that DM may play a fundamental role not only in healthy brain function and cognition, but also in determining which cognitive domains, and to what extent, are affected by age-related cognitive decline. As a potential cognitive consequence of the age-related reduction in the ability to suppress DM activity, older people might be disproportionately more distracted by task-irrelevant demands[49].

In this study, we focused primarily on task-evoked responsivity (activation or suppression) of functional brain components to cognitive ageing. In contrast, a number of recent studies have investigated changes in FC, showing that connectivity within and between brain networks, such as task-positive and DM networks, relate to behavioural effects of ageing[24,50,51]. Age-related differences in component responsivity and network coupling are likely inter-related, potentially with one process causing the other or both driven by some underlying cause. We thus suggest that the investigation of the interaction of these processes will be essential in furthering our understanding of the neural mediators of cognitive ageing.

In summary, our results provide evidence across domains for successful cognition carried out by the orchestrated activation of multiple, spatially distributed task-specific (task-positive) components, the identity of which is remarkably stable across the adult lifespan. Importantly, the cognitively declining tasks were associated with an age-related decrease in task-positive

## Table 1 | Participant demographics and MMSE score.

| Age group | Young | Middle | Older | Total |
|---|---|---|---|---|
| n | 29 | 35 | 34 | 98 |
| Age range (years) | 23–45 | 46–64 | 65–87 | 23–87 |
| Sex (male/female) | 13/16 | 17/18 | 17/17 | 47/51 |
| *Highest education* | | | | |
| University | 25 | 24 | 17 | 66 |
| A' levels | 3 | 7 | 10 | 20 |
| GCSE grade | 1 | 4 | 5 | 10 |
| None over 16 | 0 | 0 | 2 | 2 |
| MMSE | 29.3 (1.2) | 29.1 (1.0) | 27.9 (1.4) | 28.8 (1.3) |

responsivity, whereas in the cognitively preserved task task-positive responsivity also did not show any age-related difference. Furthermore, while the preserved task was not associated with significant deactivation of the DM network, we found such deactivation in both declining tasks, along with an age-related decrease in DM suppression. Altogether, our results highlight the importance of maintaining high neural responsiveness for successful cognitive ageing, and suggest that age-related loss in the ability to modulate task-positive and task-negative DM activity may be one of the primary domain-specific neural causes of age-related cognitive decline.

We hope our approach and results will contribute to the strengthening of a more realistic and nuanced view of neurocognitive ageing, with potentially similar underlying neural mechanisms undergoing differential, domain-specific modulation and change over time, resulting in disparate cognitive ageing trajectories, including both decline and preservation, across cognitive domains.

## Methods

**Participants.** Participants ($n = 98$, age: 23–87, mean: 55.0, s.d.: 16.4) were recruited from the population-based sample of the Cambridge Centre for Ageing and Neuroscience (Cam-CAN) project[25] (www.cam-can.com). Participants underwent extensive cognitive, MRI and MEG testing for which they were screened by a diverse set of screening measures. Exclusion criteria included a list of significant psychiatric and health conditions (for example, history of stroke, chemo/radiotherapy, self-reported major psychiatric disorders), non-native English, poor hearing (failing to hear 35 dB at 1,000 Hz in both ears), poor vision (below 20/50 on the Snellen test), and 24 or lower MMSE score[52]. Demographic information of the current sample is provided in Table 1. Informed consent was obtained from all participants and ethical approval for the study was obtained from the Cambridgeshire 2 (now East of England—Cambridge Central) Research Ethics Committee. Only participants with complete fMRI recording from all three tasks were included in this study.

**Cognitive tasks.** Here we introduce the three tasks used in the study, including their task scores and conditions of interest. A complete description of all task condition regressors is given in Supplementary Methods. Given the strong age-related differences in some of the behavioural scores, we detected participants with outlier performance using a sliding window approach (window width around each subject: ± 15 years, outlier threshold: 2.5 s.d. within window). This resulted in the exclusion from the statistical tests (but not from the group ICA) of two participants from Fluid Intelligence (age 25 and 42), two participants from Picture Naming (age 23 and 40) and one participant from Sentence Comprehension (age 51).

The Fluid Intelligence task taps into the central cognitive process of fluid reasoning, which is believed to underlie any complex mental control program[32], and tends to show an age-related decrease[37,38]. This experiment was a simplified version of the standard Cattell Culture Fair fluid intelligence test[53], modified to be used in the scanner[54]. The task used the classification subtest of the original Cattell test, in which four patterns are presented on the screen and participants make a button press to select the odd one out. The task employed a block design, with 30 s blocks of trials alternating between two difficulty levels, easy and hard. There was a total of four blocks per condition. As the task was self-paced with no fixed set of trials attempted by every participant, we evaluated task performance using a hits-minus-misses type of measure (hard correct − hard incorrect + easy correct − easy incorrect) that effectively balances between problem solving speed and accuracy, both being important aspects of Fluid Intelligence[37]. The suitability

of this performance score was confirmed by its strong correlation (Pearson's $r$ [95% CI]: $r(96) = 0.77$ [0.64, 0.83], $P = 10^{-17}$) with scores obtained from the full version of the Cattell test (Scale 2 Form A), administered outside the scanner[25]. For the fMRI analyses, we contrasted component activity in hard (more demanding) versus easy (less demanding) blocks (see Supplementary Methods).

The Picture Naming task measures word retrieval during picture naming, which tends to show an age-related decrease[39]. At the start of each trial, a fixation point was presented for 500 ms, followed by an object for 750 ms (1,000 ms interstimulus interval). Participants were asked to name the objects out loud, as quickly as possible, and their responses were recorded (picture naming condition). All the presented objects were common with short names (one or two syllable). A total of 200 pictures were presented in a different random order for each participant. In addition to the experimental trials, participants also saw 30 trials of phase-scrambled images to which they responded with the word 'noise', to serve as a low level visuomotor baseline condition (scrambled image condition). Task performance was measured as the percentage of correctly named trials, with common synonyms accepted as correct answers. For the fMRI analysis, we contrasted component activity to correctly named objects (more demanding condition) with that to scrambled images (less demanding condition, see Supplementary Methods).

The Sentence Comprehension experiment investigates syntactic processing during online comprehension of spoken language, which tends to be preserved with age[22,23]. Syntactically ambiguous phrases with multiple grammatically valid interpretations (for example, '... landing planes...') occur frequently and naturally in spoken language, and are disambiguated by the surrounding context. Furthermore, these ambiguous phrases may be biased towards the more frequent one of their two interpretations (dominant versus subordinate), which typically requires additional syntactic processing to be overcome in the subordinate context (reinterpretion)[44]. This task aims to test the ability to reinterpret such syntactically ambiguous structures commonly occurring in spoken language. The experiment involved sentences with three levels of syntactic processing required: unambiguous sentences had only one meaning, dominant sentences contained ambiguous phrases in their more frequent meaning ('... landing planes are...'), while subordinate sentences contained ambiguous phrases with their less frequent meaning ('... landing planes is...'), requiring the most syntactic processing. During each trial, participants were presented with the first part of the sentence spoken in a woman's voice up to the end of the central phrase (for example, '... landing planes'), and after a 200-ms delay they heard the disambiguating continuation word ('is' or 'are') spoken in a man's voice. Participants were asked to decide, using a button press, whether the final word was an acceptable continuation of the sentence or not. The stimulus set consisted of 42 sentences from each sentence category (unambiguous, dominant, subordinate), and 21 auditory baseline stimuli ('musical rain' matching spectral characteristics of stimuli, not used in this study)[44]. In this task, we used reaction time (RT), rather than accuracy score, as the dependent variable, because the former are more sensitive to the graded nature of syntactical preferences than binary accuracy judgements[44]. Mean RT difference between the conditions requiring the most and the least syntactic processing, subordinate and unambiguous sentences, was used as a behavioural measure indicating processing of, and sensitivity to, syntactic structure in normal (non-pathological) subjects[22,23,55]. Anticipatory responses (<200 ms; <1% of trials) were removed from the analysis and the RTs were inverse transformed (that is, 1,000/RT before calculating condition means per subject[56]. This method reduced the influence of outlying RTs. without the loss of individual data points, which represent difficulty with overturning the dominant interpretation of the syntactic structure into a highly unexpected (but still grammatically correct) one[44]. The condition means were subsequently reverse-transformed (that is, 1,000/mean), so that higher RTs reflect slower responding. For the fMRI analysis, we contrasted component activity to subordinate sentences (more demanding condition) with that to unambiguous sentences (less demanding condition, see Supplementary Methods).

**MRI acquisition and preprocessing.** Imaging was performed on a 3T Siemens TIM Trio System at the MRC Cognition Brain and Sciences Unit, Cambridge, UK. A 3D structural MRI was acquired for each subject using T1-weighted sequence (Generalized Autocalibrating Partially Parallel Acquisition (GRAPPA); Repetition Time (TR) = 2,250 ms; Echo Time (TE) = 2.99 ms; Inversion Time (TI) = 900 ms; flip angle $\alpha = 9°$; matrix size 256 mm × 240 mm × 19 mm; field of view (FOV) = 256 mm × 240 mm × 192 mm; resolution = 1 mm isotropic; accelerated factor = 2) with acquisition time of 4 min and 32 s. For the functional runs, T2*-weighted fMRI data were acquired using a Gradient-Echo Echo-Planar Imaging (EPI) sequence (TR = 1,970 ms; TE = 30 ms; flip angle = 78°; 32 axial slices of thickness of 3.7 mm with an interslice gap of 20%; FOV = 192 mm × 192 mm; voxel-size = 3 mm × 3 mm × 4.44 mm). The lengths of the recording sessions for the different tasks were as follows: Fluid Intelligence: 5 min (150 volumes), Picture Naming: 10 min (314 volumes), Sentence Comprehension: 16 min (496 volumes).

MRI image preprocessing was carried out by the standardized pipeline of the Cam-CAN project[57]. In brief, preprocessing was performed using SPM12 (Wellcome Department of Imaging Neuroscience, University College London, London, UK), implemented in the automatic analysis batching system[58] (http://imaging.mrc-cbu.cam.ac.uk/imaging/AA). The functional images were motion-corrected (realigned) and slice-time corrected. The T1-weighted images

were coregistered to an MNI template image, bias-corrected, and segmented into various tissue classes using unified segmentation[59]. The segmented grey matter images were then used to create a study-specific anatomical template, using the DARTEL procedure to optimize inter-participant alignment[60], and was subsequently transformed to MNI space. The EPI images were then coregistered to the T1 image, normalized to MNI space using the DARTEL flowfields, and smoothed using an 8 mm FWHM Gaussian kernel.

**Data cleaning.** A schematic diagram of the data processing steps are shown in Supplementary Fig. 1. After standard preprocessing (see previous section), we applied additional data cleaning steps to remove artefacts originating from in-scanner head motion[61], using the Automatic Removal of Motion Artifacts software package[62] (ICA-AROMA, see Data Availability). Following the pipeline used by the authors of ICA-AROMA, we subsequently regressed out the mean WM and cerebrospinal fluid (CSF) signal from the time-course of each voxel, and linearly detrended the residual signal. No temporal filtering was applied on the data, because ICA can benefit from information from the full range of the frequency spectrum in separating the signal sources, and the spectral properties of the obtained components can be subsequently used to identify and exclude those with a high proportion of signal power at non-neural (for example, vascular) frequency bands (see next section, Independent Components Analysis).

**Independent components analysis.** Spatial ICA identifies functional brain components by decomposing the 4D fMRI image into a spatially maximally independent set of signal sources (components), each associated with a spatial map (location) and a time-course (activity)[26,63]. The preprocessed and cleaned fMRI data (see previous sections) from all three tasks were submitted to the same ICA, using the Group ICA of fMRI Toolbox[63] (GIFT, see section Code availability), in order to derive a common set of components across the different tasks.

There are a number of points to consider when running an ICA on multiple tasks (rather than on a single task). On the one hand, as the ICA is forced to find a single consensus partitioning across all tasks, tasks with different signal sources may undesirably alter each other's task-specific components. On the other hand, however, at an ideal resolution of decomposition one would expect to find a shared, but differentially engaged, set of components (functional areas) across all tasks. With these points in mind, we determined the optimal number of components for our data as the resolution yielding the highest convergence between single-task and multiple-task ICAs, more specifically, the resolution at which the spatial overlap between the single-task and multiple-task ICA components, on average across all tasks, is the largest. We found this optimal convergence to occur at $n = 50$ components (see Supplementary Methods and Supplementary Fig. 2). Accordingly, an $n = 50$ component ICA was run across all three tasks temporally concatenated (as multiple sessions in GIFT) and across all subjects. The ICASSO module with 100 repetitions was used to estimate robust group level components, which were then back-reconstructed to yield task- and subject-specific time-courses (using the GICA3 method, see Supplementary Fig. 1).

We used a mixture model method[16] to threshold each component's spatial map. Potential WM- and CSF-related components were tested by calculating the loading-value-weighted overlap of the components' thresholded spatial maps with segmented tissue masks (see section MRI Acquisition and Preprocessing). Owing to the extensive data cleaning (see section Data cleaning), we found no obvious motion-, WM- or CSF-related components.

Subsequently, components related to vascular and noise-related activity were identified based on a combination of spatial and spectral criteria. Specifically, components were marked as non-neural if either their time-course exhibited below threshold low-to-high frequency spectral power ratio (<0.65) or dynamic range (<0.0175), both calculated by GIFT[16], or less than 75% spatial overlap with the cohort-specific grey matter mask. This procedure identified 17 components exhibiting strongly non-neural signal characteristics, all of inferio- or subcortical origin spatially, many of them well-matching major pathways of the venous and artery systems of the brain. These components were excluded from further analysis, resulting in 33 components of cortical origin (see Supplementary Figs 3–7). Finally, the remaining components of neural origin were labelled using their weighted spatial overlap with the Harvard–Oxford anatomical atlas[64] and a resting-state functional network atlas[16].

**Component responsivity.** Functional modulation (activation or suppression) of each component for each subject and task was estimated using standard multiple regression analysis, with task conditions as predictors and subject-specific component time-course as dependent variable, yielding standardized $\beta$ loading values (see Supplementary Methods). Then, functional responsivity of each component in each task was calculated as the difference between the $\beta$ values of the task's two conditions of interest (see section Cognitive Tasks). Considering the nature of the selected conditions, the responsivity values represent the amount of excess activation/suppression of the component from the task's less demanding condition to the more demanding one for each subject.

Finally, we excluded from the analysis three further components corresponding to the primary motor areas: left and right dorsal motor cortex, activated for button press responses in Fluid Intelligence and Sentence Comprehension, and bilateral

ventral motor cortex, activated for speech responses in Picture Naming (see Supplementary Fig. 3a–c). Although these components are very strongly activated and related to performance score in certain tasks (where higher task scores correlate with more responses given, yielding higher responsivity in the relevant motor cortex region), they are not likely to be related to core cognitive processes under investigation, and thus could be a source of confound. This final exclusion step decreased the number of brain components of interest to $n = 30$.

**Component responsivity and task performance.** We tested the hypothesis that (cohort-mean) component responsivity is a proxy for the component's contribution to task execution, that is, that more responsive components are more related to task performance than less responsive ones. To this end, for each component and task, we first calculated the correlation between task performance and component responsivity across all participants, as an index of the component's contribution to performing the task. Then, for each task, we tested whether more responsive components are also contributing more to task performance by correlating, across components, the cohort-mean responsivity of the components and their relation to performance.

**Multi-component and task-positive responsivity.** We measured the collective responsivity of a given set of components (multiple component responsivity, MCR) by taking the mean over the components' responsivities, for a specific participant and task. Then, for each task, we defined the task-positive components as the sets of components whose MCR is the most predictive of task performance across participants. In order to narrow down the search for these task-positive components among all the possible sets of components ($n = 2^{30} \approx 1.1$ billion sets for 30 components), we applied the heuristic that the most responsive components are likely to contribute most to task performance (see Fig. 2), and tested only the sets of components comprised of the first $k$ most responsive components ($1 \le k \le 30$, $n = 30$ tests per task). More specifically, we correlated each task's MCR to task performance iteratively, starting by correlating performance to the single most responsive component, and incorporating the next most responsive component with each iteration. Finally, we defined MTR as the MCR calculated on the task-positive components, which provided a participant-specific metric of neural responsiveness to each task.

**Functional connectivity.** We estimated condition-specific FC between each pair of components for each task and subject. First, realignment parameters, their derivatives, squared terms and squared derivatives were regressed out of the component time-courses, in order to further decrease artefactual coupling between component time-courses induced by in-scanner head-motion[65]. Second, the canonical haemodynamic response function convolved time-courses of all the conditions of non-interest were also regressed out of the residual component time-courses in order to minimize their impact on the connectivity. Third, condition-specific FC was calculated as the Pearson correlation coefficient of each component-pair's activity restricted to the time-points (volumes) when at least one of the two conditions of interest was present (their haemodynamic response function convolved time-course was greater than 0.05). The obtained correlation values were subsequently Fisher $z$-transformed ($z = \text{atanh}(r)$) to normalize the distribution of correlation values. We note that using standard FCs from the entire recording, rather than rendering them condition specific (with steps two and three above), provided qualitatively the same results that we report using the condition specific FCs.

**Statistical analysis.** Covariates and correlation tests: To minimize potential cohort or generation effects in our cross-sectional sample[12], we controlled for level of education, along with handedness score and gender in all statistical tests, by regressing them out from the variables of interest and performing the statistical tests on the regression residuals. Most statistical tests were run across the entire cohort, using age as a continuous variable, except for some confirmatory tests. In these latter tests, we split the cohort into three, approximately equal-size age groups (younger, middle-aged and older, see Table 1) to test for effects in the different age-groups separately.

For each correlation test, we report Pearson's correlation coefficient $r$ (and $p$) values, and 95% confidence intervals (CI). To test for significant difference between correlations coefficients, we calculated the $z$-value of the difference between the Fisher $z$-transformed correlation coefficients (using the paired.r function of the psych software package[66]), while correcting for potential sample size differences and inter-dependence between the correlations.

Moderation and mediation tests: We used multiple linear regression with interaction to test the potential moderation effect of a variable on the relation between two other variables[67]. More specifically, if $X$ and $Y$ are the variables forming the original relationship, and $Z$ is the putative moderator variable to be tested, we ran a multiple linear regression with $Y$ as the dependent variable, and $X$, $Z$ and the interaction term $X \times Z$ as predictor variables (along with our standard covariates, see first paragraph of section). A significantly non-zero coefficient of predictor $X \times Z$ would in turn indicate a moderator effect of $Z$ on the relation between $X$ and $Y$.

We performed mediation analyses[67] using the R statistical package 'mediation'[68]. We note that mediation models, like any other current statistical models, are unable to delineate time-dependent relations and causal structure in cross-sectional data sets[45]. However, when interpreted with caution, they are capable of representing age-related differences and variance partitioning[69], making them an informative complementary method to standard correlation and regression tests.

**Data availability.** The data set analysed in this study is part of the Cambridge Centre for Ageing and Neuroscience (Cam-CAN) research project (www.cam-can.com). The entire Cam-CAN dataset is soon to be publicly released, and will be available after registration via the Cam-CAN dataset inventory at https://camcan-archive.mrc-cbu.cam.ac.uk/dataaccess/.

Data cleaning was performed using the ICA-AROMA package, freely downloadable from https://github.com/rhr-pruim/ICA-AROMA. To run group ICA, we used the Matlab package GIFT, version 4.0, which is freely available at: http://mialab.mrn.org/software/gift/index.html. Mediation analyses were run using the R package 'mediation', version 4.4.5, available freely at: https://cran.r-project.org/web/packages/mediation/index.html. All surface rendered images were generated by the Matlab package BrainNetViewer, version 1.53, freely available at: https://www.nitrc.org/projects/bnv/. The corresponding author (D.S.) can provide custom-written analysis code on request.

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

## Acknowledgements

The Cambridge Centre for Ageing and Neuroscience (Cam-CAN) research was supported by the Biotechnology and Biological Sciences Research Council (grant number BB/H008217/1). K.A.T. is supported by Wellcome Trust (RG73750-RRZA/040) and British Academy Postdoctoral Fellowship (PF160048). We are grateful to the Cam-CAN respondents and their primary care teams in Cambridge for their participation in this study. We also thank colleagues at the MRC Cognition and Brain Sciences Unit MEG and MRI facilities for their assistance.

## Author contributions

D.S., K.L.C., K.A.T., M.A.S. Cam-CAN and L.K.T. devised experiment, Cam-CAN collected data, D.S. performed analysis and created figures, D.S., K.L.C., K.A.T., M.A.S. and L.K.T. wrote the paper.

## Additional information

**Competing interests:** The authors declare no competing financial interests.

**Cam-CAN consortium**

Carol Brayne[2], Edward T Bullmore[2], Andrew C Calder[2], Rhodri Cusack[2], Tim Dalgleish[2], John Duncan[2], Richard N Henson[2], Fiona E Matthews[2], William D Marslen-Wilson[2], James B Rowe[2], Teresa Cheung[2], Simon Davis[2], Linda Geerligs[2], Rogier Kievit[2], Anna McCarrey[2], Abdur Mustafa[2], Darren Price[2], Jason R Taylor[2], Matthias Treder[2], Janna van Belle[2], Nitin Williams[2], Lauren Bates[2], Tina Emery[2], Sharon Erzinçlioglu[2], Andrew Gadie[2], Sofia Gerbase[2], Stanimira Georgieva[2], Claire Hanley[2], Beth Parkin[2], David Troy[2], Tibor Auer[2], Marta Correia[2], Lu Gao[2], Emma Green[2], Rafael Henriques[2], Jodie Allen[2], Gillian Amery[2], Liana Amunts[2], Anne Barcroft[2], Amanda Castle[2], Cheryl Dias[2], Jonathan Dowrick[2], Melissa Fair[2], Hayley Fisher[2], Anna Goulding[2], Adarsh Grewal[2], Geoff Hale[2], Andrew Hilton[2], Frances Johnson[2], Patricia Johnston[2], Thea Kavanagh-Williamson[2], Magdalena Kwasniewska[2], Alison McMinn[2], Kim Norman[2], Jessica Penrose[2], Fiona Roby[2], Diane Rowland[2], John Sargeant[2], Maggie Squire[2], Beth Stevens[2], Aldabra Stoddart[2], Cheryl Stone[2], Tracy Thompson[2], Ozlem Yazlik[2], Dan Barnes[2], Marie Dixon[2], Jaya Hillman[2], Joanne Mitchell[2], Laura Villis[2]

