## [Peer Review File · Nature Communications]

Reviewers' comments:

Reviewer #1 (Remarks to the Author):

In this paper the authors report fMRI data from 98 adults aged 23-87, who performed three different tasks: a fluid intelligence task, a picture naming task, and a sentence comprehension task. Behaviorally the authors report age-related declines on the fluid intelligence and picture naming tasks (accuracy), but not the sentence comprehension task (RT). The authors then use ICA to examine the relationship between brain networks and task performance and report a host of analyses related to component responsiveness, age, and performance. They find correlations between task-positive activation and behavioral performance for the fluid intelligence and picture naming tasks, but not the sentence comprehension task. In all they interpret their findings as supporting a maintenance view of aging, with better performing older adults showing patterns more similar to those of young adults.

Overall, there is much to like about this paper, which is a useful contribution to our understanding of the cognitive aging process. The sample is admirable, and the overall analysis approach appropriate (and Herculean). However, I had a number of methodological and theoretical questions and concerns that tempered my enthusiasm.

COMMENTS TO AUTHORS

Note: To improve the transparency of peer review and editorial decisions, I avoid entering anything in boxes like the "Confidential Comments to the Editor", partly out of concern that such boxes can easily be abused with "stealth rejections" and/or unsupported claims. (In this case I apologized for my tardy review.) I'm happy to see that Nature Communications doesn't have any other hidden fields that are not shared with the authors.

MAJOR COMMENTS

-I am sympathetic to your characterization of online syntactic processing being preserved in older age and recognize there are differing opinions on this topic. Given that the sentence comprehension measure is RT, doesn't this count as an offline/post-perceptual measure? I guess I would expect to see older adults taking differentially longer on this task. I had trouble understanding the RT data (see below) but at least some comment on this issue would be welcome, as many studies find increased RTs and/or increased brain activity in older adults on syntactically complex sentences (at least when an offline task is required). It seems important not to oversimplify this point.

-The "Task scores" in Figure 1 and the results are opaque and should be clarified. This could easily be done by labeling the y axes appropriately. When I look at the methods, I'm still

slightly confused. For example, on p. 23 you note that the fluid intelligence test is evaluated as mean d-prime, but the y axis in Figure 1 is in the range of 50-90. It's also not clear whether the values in Figure 1 reflect overall performance or difference between easy and hard blocks. For picture naming, the methods on p. 24 say the measure is proportion of correctly named trials; presumably this is converted to percent in Figure 1, but this is not noted. For sentence comprehension, the score is the difference score between two conditions, but why these two conditions, and why a difference score in ms, is not explained.

-The use of accuracy rather than latency scores for fluid intelligence and picture naming is understandable, but contrasts with the RT measure for sentence comprehension. At the very least, the supplemental materials need to show the accuracy and RT for all 3 tasks (for every condition), and the use of one vs. the other measure justified. This is an important point as it speaks to the core of your argument about preservation vs. decline.

-Given that RTs are used for the sentence comprehension task, some greater consideration of how the RTs are calculated is warranted. RT analyses can be complicated generally, and even more challenging in aging when there is typically an overall age-related slowing of response times. I understand using a difference score is probably intended to counter this, but I'm not sure it's the best approach: difference scores can be more variable because they represent the difference between two estimated quantities. As far as I know the most accurate approach to examining RTs in the context of group (or age) differences is to Z-transform scores within each participant across all conditions (after trimming outliers, which I don't think was done in the current analysis - e.g., 2.5 SD from the mean RT per condition), and then use these scores in analyses. More attention to the RT analysis is warranted given the claim that it shows no age effects. One example of this approach is Faust et al (1999).

-Performing ICA over all participants and tasks makes sense. However, doesn't this assume that components are similar in young and older adults? If young and older adults actually had different components (i.e. different patterns of connectivity), I think it would be obscured by the current analysis, would it not? Some comment on this issue would be welcome.

-Making the stimuli and data publicly available (e.g., following guidelines at <http://openessinitiative.org>) would be an excellent step. For example, using Open Science Framework (<http://osf.io>). See also <http://centerforopenscience.org/top/>. If this is not possible, please include the reason why in the paper so it becomes part of the scientific record.

-Please consider uploading unthresholded statistical maps to <http://neurovault.org> (or a similar repository). Doing so enables readers to get a richer sense of the dataset, and facilitates future meta-analyses.

-In addition to (or instead of) p values, please provide effect size estimates and 95% confidence intervals (or a Bayesian equivalent) for all behavioral results (e.g., Cohen 1994;

Cumming, 2014; Morey et al., 2016).

MINOR COMMENTS

-Please consider including figures in the text of the manuscript, rather than at the end. Inline figures make the manuscript much easier for me to read!

-p. 7 The Pearson r values given in the text don't all agree with the r values given in Figure 1.

-p. 22 The windowed approach to outlier rejection seems reasonable. How was this implemented at the ends of the age range? Did those windows contain fewer than 30 years of data? (i.e. for the 23 year old, you would exclude based on 23-38; for the 40 year old, 25-55)?

-Please include all 3 sentence types and musical rain in the plots for sentence comprehension (at least in supplemental material). (In fact i'd like to see activity for all conditions for all tasks, but I think sentence comprehension is the only one missing a condition.)

-I apologize if I missed this in the methods, but how were across-subject differences in accuracy controlled for? In a task-related GLM one could simply analyze correct trials only. I'm wondering about this in the context of higher-performing older adults showing more similar patterns to the young adults - don't they also have more similar behavior? This seems like a potential confound.

-It seems you used a continuous scanning sequence for the auditory sentence comprehension task, meaning the sentences were presented in the presence of acoustic scanner noise. Although I understand there are reasons for doing so, this point should be mentioned explicitly, as it almost certainly changes the cognitive demands on the listeners (and may interact with age). (Possibly supported by the anterior cingulate activity coming out with the auditory cortex component in Supplementary Figure 3.)

-p. 30 Notes the calculation of component grey matter concentration, but I couldn't find this referred to in the results. (I had some questions about how this was calculated but if it's not reported in the results those don't matter.)

NO RESPONSE NEEDED

-The study and analyses are ambitious and no doubt difficult to present parsimoniously. That being said, I had trouble following the many different analyses, not all of which are common, particularly given the necessary splitting of the methods section after the results. Anything that could be done to help improve the flow, or to include some more relevant

methodological details in the results, would be useful. I realize this may not be possible, but as a reader I struggled to wade through the results (after several cups of coffee).

References:

Cohen J (1994) The earth is round ($p < .05$). *American Psychologist* 49:997-1003.

Cumming G (2014) The new statistics: Why and how. *Psychological Science* 25:7-29.

Drummond GB, Vowler SL (2011) Show the data, don't conceal them. *Journal of Physiology* 589.8:1861-1863. <http://dx.doi.org/10.1113/jphysiol.2011.205062>

Faust ME, Balota DA, Spieler DH, Ferraro FR (1999) Individual differences in information-processing rate and amount: Implications for group differences in response latency. *Psychol Bull* 125:777-799.

Morey RD, Hoekstra R, Rouder JN, Lee MD, Wagenmakers E-J (2016) The fallacy of placing confidence in confidence intervals. *Psychonomic Bulletin and Review* 23:103-123. <https://learnbayes.org/papers/confidenceIntervalsFallacy/>

Reviewer #2 (Remarks to the Author):

The authors report analyses of cognitive performance and functional brain activity in three tasks performed by an age-heterogeneous (23-87 years; $N = 98$) cross-sectional sample of individuals. Results show that between-person differences in task-positive brain activations predict differences in performance, age-differences in cognitive performance are to some degree shared with ("mediated by") individual differences in task-positive activations, no evidence for higher task-positive brain activations in older adults, and suppression of default mode activity during the age-sensitive task that also is reduced in older adults. I have two major concerns that stop me from recommending publication of this paper.

The mediation models are not described in the statistical procedures, but more importantly, such analyses of age-heterogeneous cross-sectional data have been rightfully criticized. I will not repeat the critique here, but refer to the nice overview by Lindenberger et al. (2011, *Psychology and Aging*). In essence, these analyses do not allow for examining whether the aging of performance is associated with aging of the functional activity. Assuming no age-related selectivity, higher correlation between the domains in the older adults than in the younger could possibly do so (Hofer et al., 2006, *Multivariate Behavioral Research*), but results here suggest no age-moderation of this kind. With this in mind, I find that the manuscript adds little to the understanding of functional correlates of cognitive aging. On a related side note, given the absence of longitudinal data, the authors must generally tone

down the language related to how informative the results are of aging. For example, "aging patterns" can clearly only be detected with longitudinal data: In the older sample here, how do you know whether the activity of a person has been in the observed way since age 20 or whether it reflects a change during the person's aging process? This is only one example.

With this in mind, the findings that task-positive brain activations predict differences in performance, that older adults do not functionally over-recruit, and that suppression of default mode activity occur in some tasks and are reduced in older adults remains. Though these effects are demonstrated with elegant analyses and a large sample, they are hardly novel. Similar findings have been previously reported by several groups (e.g., Nagel et al., 2009, PNAS; Park and Reuter-Lorenz, 2009, Annual Review of Psychology; Persson et al., 2007, Journal of Cognitive Neuroscience). I therefore find that these results are more appropriately reported in a specialty journal.

Reviewer #3 (Remarks to the Author):

Manuscript Summary

The authors investigate whether and how functional brain changes mediate age-related cognitive change, and whether the pattern of brain changes is consistent with compensation versus maintenance accounts of neurocognitive aging. Using ICA methods to identify task-based network components, they report that responsivity in the most active components is predictive of task performance (and decline) for tasks known to decline with age (i.e. fluid intelligence, picture naming) but not those that are robust to decline across the lifespan (i.e. sentence completion). Further, reduced default network suppression predicted age-related change on the two declining tasks but not the stable task. The authors conclude that functional brain changes mediate the relationship between aging and cognitive decline, and preservation of 'younger' patterns of brain activity in older adulthood is predictive of better cognitive aging, consistent with a maintenance account.

Overall review

This study addresses a critical, and surprisingly, poorly understood question of how functional brain activity mediates cognitive changes across the adult lifespan. The study design is clever, including tasks that decline and remain stable with advancing age. The literature review situates the work within the broader context of neurocognitive aging theory and provides a clear rationale for the study. The methodological approach is sufficiently detailed to allow for replication, and the presentation of the results is generally clear and transparent. The conclusions are fully supported by the data, although the discussion could be strengthened by referencing more recent findings characterizing network changes and cognitive aging. A few substantive methodological issues require further clarification, and a more thorough discussion of the findings would strengthen the impact of the paper. These points are discussed briefly below.

1. If I understand the methodological approach correctly, a core neural metric in the paper - functional responsivity - is dependent on modulation of brain activity from easier to more difficult conditions within each task. However, this is only described in the supplemental data (page 14) and should be more clearly detailed in the main methods. Further, given the centrality of the difficulty manipulations, very few behavioral findings are presented in the paper or the supplemental results. Was the difficulty manipulation successful for each task, and across age? If not - or if there were age-related differences in the within-task manipulations, how might this impact the interpretation of the brain data vis-à-vis functional compensation? On a related point, for all of the task performance correlations - what was the specific behavioral variable used in these calculations (i.e. performance on the easy/difficult condition, or performance differences between conditions)? This is unclear in the text and the Figures.

2. While the benefits and risks of conducting the multi-task ICA are briefly reviewed in the paper - it is unclear that the choice of single vs. multi-component spatial mapping to select the final group component architecture is an optimal approach. By constraining the group components with the task-specific component maps is there a bias against detection of cross-task components? Given that the cross-task ICA is a foundation for the paper, further discussion of the rationale and potential risks/biases should be more clearly laid out in the main text. This issue could also be addressed by supplementing the current data-driven approach with analyses using standard univariate methods, or using well-validated functional network models (and corresponding ROIs) referenced in the current paper (Yeo et al., Power et al.).

3. The components included in the functional connectivity analyses are quite spatially extensive, likely including diverse functional zones (e.g. right and left IPL components of the DN almost certainly include nodes of the canonical frontal-parietal control network). This makes it difficult to interpret the functional connectivity findings as nodes of several brain networks may be included or intermixed among the components in the current analysis. As suggested in point 2, perhaps replicating these analyses using more spatially constrained ROIs would help to demonstrate the robustness of the current findings.

4. The definition of functional compensation as a 'positive correlation between increased brain activity and cognitive performance' seems somewhat limited given recent literature suggesting that altered network dynamics - not simply magnitude of activation changes - may be a critical component of functional compensation (e.g. see recent work from Geerligs and colleagues, 2012 or Spreng and Schacter, 2012). While altered network dynamics is investigated here, the link to the compensatory hypothesis of aging isn't fully developed. This is particularly evident in the discussion of the finding that domain general components appear to demonstrate context-specific functional preservation. There is emerging evidence that magnitude of activation is a poor predictor of cognitive aging and that altered network coupling - particularly involving default network regions (as demonstrated here) may be a crucial mediator of age-related cognitive decline (e.g. Turner and Spreng, 2015). Further elaboration of this network modulation aspect of functional compensation is necessary.

On a related note, the link between the current findings and the neural variability account of

cognitive aging proposed by Garret and colleagues made in the discussion (pages 20-21) is unclear. The authors state that their findings are consistent with and extend this account - however, the explanation doesn't fully elaborate on why this is the case. While the neural variability account of cognitive aging is an important and valuable new theoretical approach, the relationship to the current findings is not sufficiently elaborated.

5. Overall, the Figure captions require more detail. For example, what do the color variations in Figure 5, Panel A mean? Figure 2 is quite difficult to interpret as it represents correlations of correlations for each component. This may be the optimal way of representing these findings but it was challenging to interpret the main results being conveyed in the Figure. Perhaps more detail in the Figure caption could help.

Reviewer #1 (Remarks to the Author):

In this paper the authors report fMRI data from 98 adults aged 23-87, who performed three different tasks: a fluid intelligence task, a picture naming task, and a sentence comprehension task. Behaviorally the authors report age-related declines on the fluid intelligence and picture naming tasks (accuracy), but not the sentence comprehension task (RT). The authors then use ICA to examine the relationship between brain networks and task performance and report a host of analyses related to component responsiveness, age, and performance. They find correlations between task-positive activation and behavioral performance for the fluid intelligence and picture naming tasks, but not the sentence comprehension task. In all they interpret their findings as supporting a maintenance view of aging, with better performing older adults showing patterns more similar to those of young adults.

Overall, there is much to like about this paper, which is a useful contribution to our understanding of the cognitive aging process. The sample is admirable, and the overall analysis approach appropriate (and Herculean). However, I had a number of methodological and theoretical questions and concerns that tempered my enthusiasm.

COMMENTS TO AUTHORS

Note: To improve the transparency of peer review and editorial decisions, I avoid entering anything in boxes like the "Confidential Comments to the Editor", partly out of concern that such boxes can easily be abused with "stealth rejections" and/or unsupported claims. (In this case I apologized for my tardy review.) I'm happy to see that Nature Communications doesn't have any other hidden fields that are not shared with the authors.

MAJOR COMMENTS

-I am sympathetic to your characterization of online syntactic processing being preserved in older age and recognize there are differing opinions on this topic. Given that the sentence comprehension measure is RT, doesn't this count as an offline/post-perceptual measure? I guess I would expect to see older adults taking differentially longer on this task. I had trouble understanding the RT data (see below) but at least some comment on this issue would be welcome, as many studies find increased RTs and/or increased brain activity in older adults on syntactically complex sentences (at least when an offline task is required). It seems important not to oversimplify this point.

We thank the reviewer for raising this point on the nature of our Sentence Comprehension task. Whether or not a specific behavioural measure is an on-line measure depends on when (with what latency) it is collected in relation to the variable of interest. In language comprehension studies, for example, if the variable of interest is somewhere in the middle of the sentence, and the RT is collected at the end of the sentence, that would not be an on-line measure. In contrast, if the RT is collected immediately after the variable of interest, that in our view is a more on-line measure in the sense of capturing the subject's immediate response to the key variable. In our study the subjects make their response straight after the variable of interest, i.e. after they hear the disambiguating word, making RT an on-line measure. So our Syntactic Comprehension task differs strongly from those typically used in the syntactic complexity literature that the reviewer refers to. These latter tasks, in which people typically make their judgements at the end of long, complex sentences, employ offline, more memory-dependent measures.

We have now added a table with detailed behavioural results (Supplementary Table 1), showing that overall mean reaction time (RT) per condition increased with age, as one might expect. However, the difference between conditions, which is the critical measure of syntactic processing here (see Tyler et al., 2013; Campbell et al., 2016, and also our answer to the reviewer's major comment #3 below), did not differ.

-The "Task scores" in Figure 1 and the results are opaque and should be clarified. This could easily be

done by labeling the y axes appropriately. When I look at the methods, I'm still slightly confused. For example, on p. 23 you note that the fluid intelligence test is evaluated as mean d-prime, but the y axis in Figure 1 is in the range of 50-90. It's also not clear whether the values in Figure 1 reflect overall performance or difference between easy and hard blocks. For picture naming, the methods on p. 24 say the measure is proportion of correctly named trials; presumably this is converted to percent in Figure 1, but this is not noted. For sentence comprehension, the score is the difference score between two conditions, but why these two conditions, and why a difference score in ms, is not explained.

We thank the reviewer for pointing out these omissions on how we calculated and present the task scores. As suggested, we have added labels to Figure 1 specifying the type of behavioural score used for each task.

We now clarified the calculation of Fluid Intelligence scores (page 27, 1st paragraph), making it clear that it is a “hits minus misses” type of score (correct – incorrect responses), but not actually a d-prime score (no z-scoring was applied). Conceptually, this measure better represents both the accuracy and speed aspects of the task, which are both important features of Fluid Intelligence. To verify this point empirically, we report in the main text that this score is highly correlated with the full paper-and-pencil version of the Cattell test given under standard testing conditions ($r=0.77$) (page 27, 1st paragraph).

Also, the Methods now correctly say that the Picture Naming score is the percentage (rather than proportion) of correctly named trials (page 27, 2nd paragraph).

Finally, we have now added clarification to Methods as to why we used subordinate and unambiguous sentences to calculate the behavioural score in Sentence Comprehension (page 28, 1st paragraph). Briefly, subordinate and unambiguous sentence types were chosen because these are the conditions requiring the most and the least amount of syntactic processing, respectively (Tyler et al., 2013; Campbell et al., 2016, please see also our reply to the next point).

-The use of accuracy rather than latency scores for fluid intelligence and picture naming is understandable, but contrasts with the RT measure for sentence comprehension. At the very least, the supplemental materials need to show the accuracy and RT for all 3 tasks (for every condition), and the use of one vs. the other measure justified. This is an important point as it speaks to the core of your argument about preservation vs. decline.

For each task, our aim was to select the behavioural measure which most closely reflects the cognitive process under investigation. Whether that measure is based on RTs or accuracy depends on the task in question, as is common in most cognitive research. For instance, accuracy scores are virtually meaningless in

a priming study designed to use RT measures, and RT measures are virtually meaningless in a free recall test designed to measure memory of previously studied material.

The measures used here are based on previous research using these tasks (Campbell et al. 2016; Davis et al., 2014; Shafto et al., 2010; Tyler et al., 2011; Tyler et al., 2013; Woolgar et al., 2013) and are the most appropriate to measure the cognitive function of interest in each instance. The Cattell test of fluid intelligence historically measures accuracy or number of problems solved out of those attempted within a set period of time. The Picture Naming task is designed to measure verbal production, a sort of proxy measure for the production errors commonly referred to as tip-of-the-tongue states, and accuracy is again the best measure of this process.

The situation of the Sentence Comprehension task, however, is completely different. While acceptability and RT scores show a similar pattern across age (now reported in Supplementary Table 1), acceptability scores also show substantial ceiling effects in all age-groups, as expected from this normal, non-pathological population (Tyler et al., 2011). Accuracy scores in the Sentence Comprehension task therefore are less sensitive to individual differences in syntactic processing. As opposed to this, RTs are a more sensitive measure in Sentence Comprehension, because even if someone accepted most sentences, they likely made slower decisions to the subordinate sentences, which require revision of their initial interpretation. This latter phenomenon, syntactic sensitivity, is the cognitive process we intended to measure with this task, therefore we believe that RTs are both the conceptually more appropriate and practically more sensitive type of score to use for this task. We now make this point explicit in the Methods (page 30, 2nd paragraph)

-Given that RTs are used for the sentence comprehension task, some greater consideration of how the RTs are calculated is warranted. RT analyses can be complicated generally, and even more challenging in aging when there is typically an overall age-related slowing of response times. I understand using a difference score is probably intended to counter this, but I'm not sure it's the best approach: difference scores can be more variable because they represent the difference between two estimated quantities. As far as I know the most accurate approach to examining RTs in the context of group (or age) differences is to Z-transform scores within each participant across all conditions (after trimming outliers, which I don't think was done in the current analysis - e.g., 2.5 SD from the mean RT per condition), and then use these scores in analyses. More attention to the RT analysis is warranted given the claim that it shows no age effects. One example of this approach is Faust et al (1999).

This was an omission on our part and is now described on page 28, 1st paragraph. For the RT data, anticipatory responses (< 200 ms) were removed (< 1% of trials) and the RTs were inverse transformed (i.e., 1000/RT) before calculating cell means per condition per subject. This method reduces the influence of outlying RTs without the loss of individual datapoints (Ratcliff 1993) and has been applied in most of our

previous work (e.g., Tyler et al., 2011; Tyler et al., 2013; Campbell et al., 2016). The condition means were subsequently reverse-transformed (i.e., $1000/\text{mean}$), so that higher RTs reflect slower responding.

-Performing ICA over all participants and tasks makes sense. However, doesn't this assume that components are similar in young and older adults? If young and older adults actually had different components (i.e. different patterns of connectivity), I think it would be obscured by the current analysis, would it not? Some comment on this issue would be welcome.

It is common practise in the group ICA literature to run a single ICA on the entire sample, not only in development/ageing populations concerned with continuous change (e.g. Allen et al., 2011; Douaud et al., 2014), but even in comparative studies using e.g. patient and control groups (e.g. Jafri et al., 2008; Lui et al., 2010). This method is extensively tested, applied and recommended in general by the authors of the ICA software package (GIFT) that we use in the current study. The rationale behind this choice is twofold: 1) in general, including more data in the ICA increases the reliability of the decomposition at the group-level as well as the ICA's estimate of representing the original data after "back-reconstruction" even on a single-subject level, 2) this approach yields a balanced but uniform set of components across the cohort that can subsequently be compared directly across participants. We extensively exploit this second property in our analysis, because it fits well with our initial aim to look at the shared neural components and how their activity change across the lifespan.

-Making the stimuli and data publicly available (e.g., following guidelines at <http://openessinitiative.org>) would be an excellent step. For example, using Open Science Framework (<http://osf.io>). See also <http://centerforopenscience>. If this is not possible, please include the reason why in the paper so it becomes part of the scientific record.

As an integral part of the CamCAN project, we are currently in the process of creating our own online database to make the CamCAN data publicly available. Parts of the dataset is already available, after a simple data user registration, at <https://camcan-archive.mrc-cbu.cam.ac.uk/dataaccess/>, and the rest will be released within the next 6 months, as required by the founding agencies of CamCAN. We added a note on this to the new Data and code availability section (page 35, 3rd paragraph).

-Please consider uploading unthresholded statistical maps to <http://neurovault.org> (or a similar repository). Doing so enables readers to get a richer sense of the dataset, and facilitates future meta-analyses.

This is a good idea, for the reasons stated, and we will upload the unthresholded ICA maps to neurovault.org in the event the paper is accepted and we can link to it.

-In addition to (or instead of) p values, please provide effect size estimates and 95% confidence intervals (or a Bayesian equivalent) for all behavioral results (e.g., Cohen 1994; Cumming, 2014; Morey et al., 2016).

We thank the reviewer for suggesting this improvement on our statistics, we have now added both effect sizes and 95% confidence intervals to all statistical tests.

MINOR COMMENTS

-Please consider including figures in the text of the manuscript, rather than at the end. Inline figures make the manuscript much easier for me to read!

Figures are now inserted into the main text.

-p. 7 The Pearson r values given in the text don't all agree with the r values given in Figure 1.

We thank the reviewer for noting this, we have now corrected this error (page 6, 2nd paragraph).

-p. 22 The windowed approach to outlier rejection seems reasonable. How was this implemented at the ends of the age range? Did those windows contain fewer than 30 years of data? (i.e. for the 23 year old, you would exclude based on 23-38; for the 40 year old, 25-55)?

In our implementation of outlier rejection the ends of the age range acted as hard stops for the sliding window, thereby keeping the width of the window constant across participants (30 years) to ensure that for each test an approximately equal number of participants was included (due to the homogeneous distribution of age in our sample). That is, using the example given by the reviewer, the outlier test of a 23 year old was based on the range of 23-53.

We note that, while this implementation was favourable in regard to keeping the sample size constant across tests, it could have resulted in the adverse effect of excluding high performing younger adults and low performing older adults from any of the declining tasks (due to the disproportionate inclusion of lower/higher

performing middle-aged adults in these tests). To rule out this possibility, we verified that this is in fact not the case: in each of the declining tasks, both excluded participants were younger than the median age (as reported on page 26, 2nd paragraph) and achieved lower than average behavioural score.

Conversely, but for similar reasons, this fixed window-size approach could have kept some low performing younger and high performing older adults undetected in the declining tasks. An examination of Figure 1 verifies that this is not the case in either of the declining tasks (there are no such outliers towards either ends of the age range).

-Please include all 3 sentence types and musical rain in the plots for sentence comprehension (at least in supplemental material). (In fact i'd like to see activity for all conditions for all tasks, but I think sentence comprehension is the only one missing a condition.)

We have now added the beta weights of the missing conditions (dominant sentences and musical rain in Sentence Comprehension) to the appropriate figures in SI (Supplementary Figures 3-7). All conditions are shown now for all tasks in these figures.

-I apologize if I missed this in the methods, but how were across-subject differences in accuracy controlled for? In a task-related GLM one could simply analyze correct trials only. I'm wondering about this in the context of higher-performing older adults showing more similar patterns to the young adults - don't they also have more similar behavior? This seems like a potential confound.

We thank the reviewer for raising this potentially important point. As we explain in Methods (page 27, 2nd paragraph), we did take only the correctly named trials in Picture Naming in calculating the functional responsivity measure. However, Picture Naming is the only task that lends itself to this kind of categorisation. Specifically, Fluid Intelligence, being a self-paced task, was implemented as a block design, rather than event-related, task. In Sentence Comprehension, the natural grammatical structures were also not intended to be classified into strict correct/incorrect categories (please see our reply to third major point above). While using all trials may indeed have made for a greater similarity between high-performing older adults and younger adults, focusing the analysis on correct trials only is unfortunately infeasible in the latter two tasks in our study due to the nature of these tasks. This reflects our intention during the design of the experiments to deliberately avoid tasks measuring strict behavioural violations, and instead used experiments that test cognitive processes in as natural task conditions as possible. We did so in order to compensate for the relative unfamiliarity some older adults may have in solving the rather artificial exercises involved in most standard experiments.

Our additional reason for not separating trials in the Fluid Intelligence and Sentence Comprehension tasks was to keep our analysis consistent with previous studies using the same tasks (Tyler et al., 2011, Woolgar et al., 2013, Davis et al., 2014, Campbell et al., 2016), thus achieving comparability with existing results.

Beyond the above reasons why separation of correct and incorrect trials is inappropriate for two of our tasks, we would like to note that it is unlikely that the lack of such separation in some tasks would severely confound our results. Specifically, 1) in accordance with the results in these tasks, we also found maintenance and the lack of compensation in the task (Picture Naming) where we did focus on correct trials only, and 2) test of (performance-related) compensation is likely to be biased towards a positive (rather than negative) finding in an analysis without such trial separation (insofar as the likely altered brain activity during incorrect trials has larger effect on the mean brain activity of low than of high performers, thereby helping separate any potential functional difference by performance). Therefore we believe that our negative finding on compensation is actually strengthened, rather than weakened or confounded, by this property of the analysis in the relevant tasks.

-It seems you used a continuous scanning sequence for the auditory sentence comprehension task, meaning the sentences were presented in the presence of acoustic scanner noise. Although I understand there are reasons for doing so, this point should be mentioned explicitly, as it almost certainly changes the cognitive demands on the listeners (and may interact with age). (Possibly supported by the anterior cingulate activity coming out with the auditory cortex component in Supplementary Figure 3.)

To ensure that both younger and older adults could hear the stimuli of Sentence Comprehension clearly, we piloted the stimuli extensively beforehand (using the same continuous scanning sequence as used here). Additionally, we performed a sound check before starting each session, and at the end of the session subjects were asked if they could hear the sentences clearly. No subjects reported being unable to hear the sentence stimuli.

If cognitive demands of older participants had been altered by scanner noise, then one would expect to find either age-related differences in behavioural performance (e.g., Wingfield et al., 2006) or possibly increased frontal activation in older adults in response to the increased cognitive demand. We found no evidence of either of these possibilities.

Following the reviewer's point on some parts of the anterior cingulate cortex (ACC) coming out with the auditory cortex component, we looked at the single-task ICA results to identify the origin of that effect (Supplementary Information, page 20, section *Single-task ICA results*). Confirming our reasoning above, we found that the auditory cortex component of the ICA on Sentence Comprehension did not contain any ACC region at all (and vice versa). Instead, we found some ACC regions being weakly mixed with the auditory

cortex component in the ICA results of the Picture Naming task, which involved verbal responses from the participants, and therefore could either represent coincidental (functionally non-significant) brain activity specific to the task, or some auditory feedback mechanisms to cognitive control.

-p. 30 Notes the calculation of component grey matter concentration, but I couldn't find this referred to in the results. (I had some questions about how this was calculated but if it's not reported in the results those don't matter.)

In our original manuscript, we did report grey matter results showing that the domain-specific task-positive networks of Sentence Comprehension, FT and ACC/AI, are not more resistant to age-related GM difference (loss) than other regions, arguing against their functional resilience due to any unique structural preservation (page 14, 2nd paragraph of original manuscript). However, due to space limitations, and to simplify the Results section (as rightly suggested by the reviewer below), we have now removed this section from Results along with the above referred section in Methods.

NO RESPONSE NEEDED

-The study and analyses are ambitious and no doubt difficult to present parsimoniously. That being said, I had trouble following the many different analyses, not all of which are common, particularly given the necessary splitting of the methods section after the results. Anything that could be done to help improve the flow, or to include some more relevant methodological details in the results, would be useful. I realize this may not be possible, but as a reader I struggled to wade through the results (after several cups of coffee).

We sympathise with the reviewer on this point. In order to improve the flow of the presentation of the results, we removed the section that we felt added the least to the central message of the paper (*Resilience of Sentence Comprehension's domain specific components*). Furthermore, to help the reader understanding the results without having to refer too much to the Methods, we added a sentence explaining our functional responsivity measure (page 8, 1st paragraph), which is the central, but non-standard, index used in the manuscript, and added further clarification to our inter-component-group functional connectivity measure (page 19, 2nd paragraph).

References:

Cohen J (1994) The earth is round ($p < .05$). American Psychologist 49:997-1003.

Cumming G (2014) *The new statistics: Why and how. Psychological Science* 25:7-29.

Drummond GB, Vowler SL (2011) *Show the data, don't conceal them. Journal of Physiology* 589.8:1861-1863. <http://dx.doi.org/10.1113/>

Faust ME, Balota DA, Spieler DH, Ferraro FR (1999) *Individual differences in information-processing rate and amount: Implications for group differences in response latency. Psychol Bull* 125:777-799.

Morey RD, Hoekstra R, Rouder JN, Lee MD, Wagenmakers E-J (2016) *The fallacy of placing confidence in confidence intervals. Psychonomic Bulletin and Review* 23:103-123. <https://learnbayes.org/papers/>

References:

Allen, Elena A., et al. "A baseline for the multivariate comparison of resting-state networks." *Frontiers in systems neuroscience* 5 (2011): 2.

Campbell, Karen L., et al. "Robust Resilience of the Frontotemporal Syntax System to Aging." *The Journal of Neuroscience* 36.19 (2016): 5214-5227.

Douaud, Gwenaëlle, et al. "A common brain network links development, aging, and vulnerability to disease." *Proceedings of the National Academy of Sciences* 111.49 (2014): 17648-17653.

Jafri, M. J., Pearlson, G. D., Stevens, M., & Calhoun, V. D. (2008). A method for functional network connectivity among spatially independent resting-state components in schizophrenia. *Neuroimage*, 39(4), 1666-1681.

Lui S, Li T, Deng W, et al. Short-term Effects of Antipsychotic Treatment on Cerebral Function in Drug-Naive First-Episode Schizophrenia Revealed by "Resting State" Functional Magnetic Resonance Imaging. *Arch Gen Psychiatry*. 2010;67(8):783-792. doi:10.1001/archgenpsychiatry.2010.84.

Shafto, M. A., Stamatakis, E. A., Tam, P. P., & Tyler, L. K. (2010). Word retrieval failures in old age: the relationship between structure and function. *Journal of Cognitive Neuroscience*, 22(7), 1530-1540.

Tyler, Lorraine K., et al. "Left inferior frontal cortex and syntax: function, structure and behaviour in patients with left hemisphere damage." *Brain* 134.2 (2011): 415-431.

Tyler, Lorraine K, et al. "Syntactic computations in the language network: characterizing dynamic network properties using representational similarity analysis." *Frontiers in psychology* 4 (2013): 271.

Wingfield, Arthur, et al. "Effects of adult aging and hearing loss on comprehension of rapid speech varying in syntactic complexity." *Journal of the American Academy of Audiology* 17.7 (2006): 487-497.

Woolgar, Alexandra, Daniel Bor, and John Duncan. "Global increase in task-related fronto-parietal activity after focal frontal lobe lesion." *Journal of cognitive neuroscience* 25.9 (2013): 1542-1552.

Reviewer #2 (Remarks to the Author):

The authors report analyses of cognitive performance and functional brain activity in three tasks performed by an age-heterogeneous (23-87 years; N = 98) cross-sectional sample of individuals. Results show that between-person differences in task-positive brain activations predict differences in performance, age-differences in cognitive performance are to some degree shared with ("mediated by") individual differences in task-positive activations, no evidence for higher task-positive brain activations in older adults, and suppression of default mode activity during the age-sensitive task that also is reduced in older adults. I have two major concerns that stop me from recommending publication of this paper.

*The mediation models are not described in the statistical procedures, but more importantly, such analyses of age-heterogeneous cross-sectional data have been rightfully criticized. I will not repeat the critique here, but refer to the nice overview by Lindenberger et al. (2011, *Psychology and Aging*). In essence, these analyses do not allow for examining whether the aging of performance is associated with aging of the functional activity. Assuming no age-related selectivity, higher correlation between the domains in the older adults than in the younger could possibly do so (Hofer et al., 2006, *Multivariate Behavioral Research*), but results here suggest no age-moderation of this kind. With this in mind, I find that the manuscript adds little to the understanding of functional correlates of cognitive aging. On a related side note, given the absence of longitudinal data, the authors must generally tone down the language related to how informative the results are of aging. For example, "aging patterns" can clearly only be detected with longitudinal data: In the older sample here, how do you know whether the activity of a person has been in the observed way since age 20 or whether it reflects a change during the persons aging process? This is only one example.*

With this in mind, the findings that task-positive brain activations predict differences in performance, that older adults do not functionally over-recruit, and that suppression of default mode activity occur in some tasks and are reduced in older adults remains. Though these effects are demonstrated with elegant analyses and a large sample, they are hardly novel. Similar findings have been previously reported by

several groups (e.g., Nagel et al., 2009, PNAS; Park and Reuter-Lorenz, 2009, Annual Review of Psychology; Persson et al., 2007, Journal of Cognitive Neuroscience). I therefore find that these results are more appropriately reported in a specialty journal.

We acknowledge the reviewer for the valuable comments and criticism. First, we would like to offer some aspects of our study to the reviewer's consideration that in our view distinguishes it from most related studies to date. Most importantly, the main distinctive feature of our study is in testing several cognitive domains (two that decline and one that is preserved) on the same set of participants using both behavioural and neuroimaging data. While the reviewer is of course right in saying that most of the findings we report have been reported before in isolation, we would like to emphasize that these studies generally relied on small, non-representative samples, and focused on age-related differences within single domains rather than taking a cross-domain approach as we did here.

Additionally, beyond our remarkably large sample size ($n=98$), our population-derived sample is also more representative than those of most studies to date, due to our epidemiologically-based recruiting methods (Shafto et al. 2014). That is, instead of relying on "opt-in" volunteers as most studies do (e.g., older adults who respond to ads in the paper, or Psychology students who participate for credit), we used an "opt-out" method – contacting people through local doctor's offices and sampling a much wider base of the population. Furthermore, our full brain data-driven analysis to obtain domain-specific and domain-general cognitive components also distinguishes our study from most standard studies focusing on predefined ROIs and brain networks.

Regarding the issues with mediation analysis on cross-sectional samples, we agree with the reviewer's point on the limitations of mediation analysis to infer causal relationships in the current context. Nevertheless, in accordance with Salthouse (2011), we believe that the analysis is still informative as a test to detect and quantify any potential shared source of variance among age, performance and brain function (i.e., that a significant portion of the age-performance relationship can be accounted for by the individuals' responsiveness), even if this cannot be interpreted as a direct (causal) effect due to the known caveats. Interpreted in this way, the results of the mediation analysis fit well with the rest of our results (stability of task-positive components with age, lack of functional compensation, evidence for the brain maintenance hypothesis, age-related decrease of default mode suppression in the declining tasks), and it is far from being the single cornerstone of our paper. Rather, it provides complementary evidence to our main thesis, that is, that differential decline in functional brain responsiveness across individuals and cognitive domains explains differences in inter-individual performance and disparate domain-specific developmental trends. Acknowledging the important concern of reviewer, we have rephrased the manuscript to make this interpretation of the mediation analysis (i.e. a test for shared source of variance, rather than causal relationship) explicit and clear (please see below).

Following the reviewer's remark, we have now added the description of the mediation analysis to Methods (page 35, 1st paragraph). To draw the reader's attention to the potential issues raised by the reviewer, we now make the limitations of the mediation analysis explicit in the relevant Methods and Results section (page 15, 1st paragraph). Additionally, in order to avoid implying unwarranted conclusions based on the mediation analysis, we now carefully rephrased all reference to these results by changing phrases such as "mediate (between X and Y)" to "account for (Z percentage of shared variance between X and Y)", and "effect of X on Y" to "relation between X and Y" throughout the manuscript.

More generally, we thank the reviewer for pointing out that some of the descriptions of the results are inappropriate in light of the limitations of cross-sectional data used in the study. We corrected these instances by changing the problematic phrases such as "ageing patterns" and "neural change" to "age-related patterns" and "age-related neural difference" in the title, abstract and throughout the main text.

References:

Salthouse, T. A. (2011). All data collection and analysis methods have limitations: reply to Rabbitt (2011) and Raz and Lindenberger (2011).

Shafto, M. A., Tyler, L. K., Dixon, M., Taylor, J. R., Rowe, J. B., Cusack, R., ... & Henson, R. N. (2014). The Cambridge Centre for Ageing and Neuroscience (Cam-CAN) study protocol: a cross-sectional, lifespan, multidisciplinary examination of healthy cognitive ageing. *BMC neurology*, 14(1), 1.

Reviewer #3 (Remarks to the Author):

Manuscript Summary

The authors investigate whether and how functional brain changes mediate age-related cognitive change, and whether the pattern of brain changes is consistent with compensation versus maintenance accounts of neurocognitive aging. Using ICA methods to identify task-based network components, they report that responsivity in the most active components is predictive of task performance (and decline) for tasks known to decline with age (i.e. fluid intelligence, picture naming) but not those that are robust to decline across the lifespan (i.e. sentence completion). Further, reduced default network suppression predicted age-related change on the two declining tasks but not the stable task. The authors conclude that functional brain changes mediate the relationship between aging and cognitive decline, and preservation of 'younger' patterns of brain activity in older adulthood is predictive of better cognitive aging, consistent

with a maintenance account.

Overall review

This study addresses a critical, and surprisingly, poorly understood question of how functional brain activity mediates cognitive changes across the adult lifespan. The study design is clever, including tasks that decline and remain stable with advancing age. The literature review situates the work within the broader context of neurocognitive aging theory and provides a clear rationale for the study. The methodological approach is sufficiently detailed to allow for replication, and the presentation of the results is generally clear and transparent. The conclusions are fully supported by the data, although the discussion could be strengthened by referencing more recent findings characterizing network changes and cognitive aging. A few substantive methodological issues require further clarification, and a more thorough discussion of the findings would strengthen the impact of the paper. These points are discussed briefly below.

1. If I understand the methodological approach correctly, a core neural metric in the paper - functional responsivity - is dependent on modulation of brain activity from easier to more difficult conditions within each task. However, this is only described in the supplemental data (page 14) and should be more clearly detailed in the main methods. Further, given the centrality of the difficulty manipulations, very few behavioral findings are presented in the paper or the supplemental results. Was the difficulty manipulation successful for each task, and across age? If not - or if there were age-related differences in the within-task manipulations, how might this impact the interpretation of the brain data vis-à-vis functional compensation? On a related point, for all of the task performance correlations - what was the specific behavioral variable used in these calculations (i.e. performance on the easy/difficult condition, or performance differences between conditions)? This is unclear in the text and the Figures.

We have now added further clarification regarding the nature of the functional responsivity measure both in Methods and Results of the main text, as suggested by the reviewer (page 8, 1st paragraph and page 32, 1st paragraph).

Further behavioural analysis and clarification on the behavioural measures used were also requested by reviewer #1 (please see their Major Points 2, 3 and 4 above). To address these important points, we added further clarification of the behavioural measures used to Methods (subsection *Cognitive Tasks*) and to Figure 1, and we now report means and standard errors per condition (both across all participants and within age subgroups) in Supplementary Table 1. As the table shows, the difficulty manipulations were successful across the age groups, both in terms of reaction time and accuracy score differences.

2. While the benefits and risks of conducting the multi-task ICA are briefly reviewed in the paper - it is unclear that the choice of single vs. multi-component spatial mapping to select the final group component architecture is an optimal approach. By constraining the group components with the task-specific component maps is there a bias against detection of cross-task components? Given that the cross-task ICA is a foundation for the paper, further discussion of the rationale and potential risks/biases should be more clearly laid out in the main text. This issue could also be addressed by supplementing the current data-driven approach with analyses using standard univariate methods, or using well-validated functional network models (and corresponding ROIs) referenced in the current paper (Yeo et al., Power et al.).

This issue of component identification and selection raised by the reviewer is indeed fundamental to the current study. The two main methods we used to this end are multi-task group ICA (to identify the full component architecture and the association between each component's activity timecourse and experimental contrasts, i.e. responsivity) and responsivity – performance analysis (to select task-positive components, Figure 3). In response to the reviewer's specific concern, we do not see how either of these methods would bias against the detection of cross-task components. On the contrary, our choice of a multi-task, rather than single-task, ICA greatly facilitated the identification of putative domain-general components, which was our original intention because it is the main interest of the study. Similarly, selecting the tasks' task-positive components on a per task basis (during the second method referred above) ensures that cross-task components are identified in a strict and unbiased way. These points are supported by our results, insofar as we obtain as many domain-general (cross-task) components (ILPFC, rLPFC, LOC, ITOC) as domain-specific components (ACC/AI, FT, LOP, rTOC) across the three tasks.

Following the reviewer's suggestion, we have now extended the Supplementary Information with results obtained from single-task ICA on each task, largely replicating our main results, and, supported by these new results, we added further discussion of the rationale and potential strengths and risks of our multi-task ICA approach (Supplementary Information, page 20, section *Single-task ICA results*).

3. The components included in the functional connectivity analyses are quite spatially extensive, likely including diverse functional zones (e.g. right and left IPL components of the DN almost certainly include nodes of the canonical frontal-parietal control network). This makes it difficult to interpret the functional connectivity findings as nodes of several brain networks may be included or intermixed among the components in the current analysis. As suggested in point 2, perhaps replicating these analyses using more spatially constrained ROIs would help to demonstrate the robustness of the current findings.

We would like to point out that the components' exact spatial extent presented on the figures is largely an issue of thresholding, which is perhaps one of the so far unresolved shortcomings of ICA and routinely done in a rather subjective manner from one study to another (although to our best knowledge other neuroimaging

analysis methods are also not immune to similar thresholding issues). Because spatial thresholding is one of the post-processing steps, including more or fewer voxels in the components' spatial maps does not change their time-courses (which are calculated in a preceding step during the ICA, please see Supplementary Figure 1), therefore the exact choice of threshold does not affect any of our findings on functional responsivity and relation to behaviour.

The spatial maps are nonetheless an informative source in locating the components of course, but with keeping the important caveat in mind that thresholding yields the false impression of strict, binary spatial extents, while some of the indicated regions are in fact more associable to the component than other. This is the reason why we included thresholded, but weighted (i.e., not binarized), component spatial maps as a compromise in the Supplementary Information, on which the relative importance of the areas of each component can be traced (Supplementary Figures 3-7.).

To address the reviewer's concern on the spatially extensive nature of the components and interpretability in comparison with publicly available atlases, we have now added additional information to Supplementary Table 2 (Supplementary Table 1 in original submission), detailing the percentage of weighted spatial overlap between our components and two previously published atlases: a resting state atlas based on a large ageing population (Allen et al., 2011) and a task-based atlas combining recordings from thousands of experiments in a meta-analysis (Laird et al., 2011).

Our left and right IPL components do indeed overlap with the fronto-parietal network, as the reviewer suggests (please see Supplementary Table 2). The activity of these components, however, strongly correlates with the canonical medial DM components both in our tasks (see Figure 6 of main text) and during resting state (see Figure 4 of Allen et al., 2011, ICs 34 and 60), and shows mixed (both positive and negative) functional connectivity with task-positive components both in our results and in resting state (Figure 4 in Allen et al., 2011). Nevertheless, we checked the robustness of our DM results against removing these lateral IPL components and keeping only the four medial DM components. We observed qualitatively the same results, with 1) significant DM suppression only in the declining tasks (mean functional responsivity and p-value of t-test: Fluid Intelligence: -0.166, Picture Naming: $r = -0.039$, both $p < 0.001$, while Sentence Comprehension: $r = -0.005$, $p = 0.39$), 2) significant segregation (negative FC) between task-positive and DM components only in the declining tasks (Fluid Intelligence: $r = -0.107$, Picture Naming: $r = -0.038$, both $p < 0.001$, while Sentence Comprehension: $r = -0.015$, $p = 0.10$), 3) age-related decline in DM suppression only in the declining tasks (Fluid Intelligence: $r = -0.55$, Picture Naming: $r = -0.37$, both $p < 0.001$, while Sentence Comprehension: $r = 0.11$, $p = 0.28$), 4) correlation between DM suppression and task-positive activation only in the declining tasks (Fluid Intelligence: $r = -0.58$, $p < 0.001$, Picture Naming: $r = -0.21$, $p = 0.04$, while Sentence Comprehension: $r = -0.08$, $p = 0.45$), 5) correlation between DM suppression and task score only in the declining tasks (Fluid Intelligence: $r = -0.44$, $p < 0.001$, Picture Naming: $r = -0.26$, $p =$

0.009, while Sentence Comprehension: $r = -0.17$, $p = 0.10$). We added a note on the results of these additional analyses to the main Results (page 21, 2nd paragraph).

4. The definition of functional compensation as a 'positive correlation between increased brain activity and cognitive performance' seems somewhat limited given recent literature suggesting that altered network dynamics - not simply magnitude of activation changes - may be a critical component of functional compensation (e.g. see recent work from Geerligs and colleagues, 2012 or Spreng and Schacter, 2012). While altered network dynamics is investigated here, the link to the compensatory hypothesis of aging isn't fully developed. This is particularly evident in the discussion of the finding that domain general components appear to demonstrate context-specific functional preservation. There is emerging evidence that magnitude of activation is a poor predictor of cognitive aging and that altered network coupling - particularly involving default network regions (as demonstrated here) may be a crucial mediator of age-related cognitive decline (e.g. Turner and Spreng, 2015). Further elaboration of this network modulation aspect of functional compensation is necessary.

We thank the reviewer for proposing these important additional analyses. Following them, we tested the potential relationship between age and the functional connectivity (FC) between DM and the task-positive (TP) components in each task, and whether age moderates the relation between FC and task score. We found a strong decrease in the anti-correlation between the DM components and the TP components only in the Fluid Intelligence task ($r = -0.58$, with a similar, $r = -0.51$ correlation if we constrain TPs to the bilateral LPFC), but no such relation in the other two tasks. This is in line with the strong negative FC between DM and TP components in (the typically higher performing) younger adults only in Fluid Intelligence task, as well as with the findings of the papers referenced by the reviewer, which used complex visuospatial planning tasks most similar to our Fluid Intelligence task.

Most crucially, however, we found neither continuous nor group-wise moderator effect of age on the relationship between task score and the FC between DM – TP in any of the tasks (not even in Fluid Intelligence), pointing to the lack of potential compensation effects. On the contrary, we found that in Fluid Intelligence, the only task in which FC correlated to performance, the same relationship holds in the older adult group as in the entire cohort (stronger segregation correlates with higher performance). This is in line with the brain maintenance hypothesis, in support of which we already report more direct evidence in the manuscript, involving both declining tasks and with higher effect size in our functional responsivity analysis (page 17, subsection *Functional maintenance*).

In sum, these analyses seem to point to functional activation and responsivity carrying more information about cognitive performance than functional connectivity at least in our dataset, despite all our efforts to clean our FC of confounding motion and metabolic artefacts, both prior ICA and during computing FC (see Methods, page 30, *Data cleaning* and page 33, *Functional connectivity*). Nevertheless, acknowledging the

likely importance of alterations in functional connectivity in neurocognitive ageing, we extended the discussion with a paragraph on the points and with the references kindly provided by the reviewer (page 25, 2nd paragraph).

On a related note, the link between the current findings and the neural variability account of cognitive aging proposed by Garret and colleagues made in the discussion (pages 20-21) is unclear. The authors state that their findings are consistent with and extend this account - however, the explanation doesn't fully elaborate on why this is the case. While the neural variability account of cognitive aging is an important and valuable new theoretical approach, the relationship to the current findings is not sufficiently elaborated.

We thank the reviewer for pointing out that this connection is unclear and due to space limitations, we have opted to remove it in favour of a discussion of the findings pointed to by the reviewer above (page 25, 2nd paragraph).

5. Overall, the Figure captions require more detail. For example, what do the color variations in Figure 5, Panel A mean? Figure 2 is quite difficult to interpret as it represents correlations of correlations for each component. This may be the optimal way of representing these findings but it was challenging to interpret the main results being conveyed in the Figure. Perhaps more detail in the Figure caption could help.

To ease the readers' interpretation, we have updated the figure labels and captions. The labels of Figure 1 are specific to the corresponding tasks. The caption and labels of Figure 2 are now more informative and we hope explain the figure more accurately. We added a clarifying note on component colours to Figure 5 caption.

References:

Allen, Elena A., et al. "A baseline for the multivariate comparison of resting-state networks." *Frontiers in systems neuroscience* 5 (2011): 2.

Laird, Angela R., et al. "Behavioral interpretations of intrinsic connectivity networks." *Journal of cognitive neuroscience* 23.12 (2011): 4022-4037.

Reviewers' comments:

Reviewer #1 (Remarks to the Author):

In their revised manuscript, the authors have adequately addressed my main concerns.

Reviewer #2 (Remarks to the Author):

My original comments cannot be fully addressed by including elaborations or limitations in the paper. The authors and I still disagree on the usefulness of reporting mediation analyses and on the novelty of the findings. It is up to the editor to make the call here.

Reviewer #3 (Remarks to the Author):

The authors have provided a thoughtful and comprehensive response to each of the points raised in my original review (R3). Indeed their additional analyses examining functional network interactivity have helped to shape my thinking with respect to network models of neurocognitive aging.

While I am satisfied with that my points have been addressed, I did note that the author's responses to two points raised by R1 (on page 4 of the Response PDF) seemed to only partially address their concerns.

First, the reviewer questioned why the decision was taken to select two tasks that use accuracy as the DV and one (that was specifically hypothesized to be distinct from the other two) that relies on RT. The authors did justify why RT is the best measure for the Sentence Comprehension task, however, they did not address or justify why the SC task was the 'best' measure of syntactic processing - i.e. why choose a task with a different DV that is known to interact with age?

Second, R1 raised a concern about the processing/analysis of RT data, specifically as it relates to group/age differences. Again, the authors now provide a full description of their processing approach, but their response didn't directly justify why their approach is optimal for this dataset.

Finally, R2 raised a concern with respect the novelty of the research, and whether it is more appropriate for a specialized journal. The authors' claim regarding the novelty of the within-subject approach is certainly valid from my reading of the literature. However, I do share R2's thoughts with respect to the relevance of this work to a broader science audience.

Reviewer #3 (Remarks to the Author):

The authors have provided a thoughtful and comprehensive response to each of the points raised in my original review (R3). Indeed their additional analyses examining functional network interactivity have helped to shape my thinking with respect to network models of neurocognitive aging.

While I am satisfied with that my points have been addressed, I did note that the author's responses to two points raised by R1 (on page 4 of the Response PDF) seemed to only partially address their concerns.

First, the reviewer questioned why the decision was taken to select two tasks that use accuracy as the DV and one (that was specifically hypothesized to be distinct from the other two) that relies on RT. The authors did justify why RT is the best measure for the Sentence Comprehension task, however, they did not address or justify why the SC task was the 'best' measure of syntactic processing - i.e. why choose a task with a different DV that is known to interact with age?

There are various reasons, based on the neurobiology of language literature, why we chose this particular type of experiment to test for ageing effects on language comprehension.

Experiments looking at syntax – arguably the core aspect of human language – typically contain complex syntactic constructions and/or violations. We chose to avoid both of these approaches because they draw heavily on domain-general processes that are more associated with performing a task and complementary off-line processes, rather than core online syntactic processing (Campbell et al., 2016; Davis et al., 2014; Grossman et al. 2002; Peelle et al., 2009; see also Shafto and Tyler., 2014 for a review). Increasing syntactic complexity, which typically involves constructing sentences that no one would actually speak, normally

activate the left IFG because of task difficulty. Similarly including violations in sentences recruits domain-general systems.

Because in the current study we are interested in everyday language comprehension abilities, we used a well-validated task that recruits the left hemisphere (LH) fronto-temporal system associated with syntactic processing. This is the only network whose grey matter integrity correlates with syntactic comprehension (Tyler et al., 2011) and where correlation between fMRI activity and syntactic comprehension has been found in LH damaged patients (Tyler et al., 2011). Moreover, syntactic comprehension abilities in patients with LH stroke are correlated with the connectivity between these two regions (Papousti et al, 2012) and damage to the white matter tracts connecting these two regions is associated with poor syntactic performance (Griffiths et al, 2012). Finally, our subsequent studies have confirmed the central role of this system in core syntactic processing in healthy subjects (Campbell et al., 2016; Davis et al., 2014).

Therefore, we used the specific linguistic manipulation of syntactic ambiguity because (a) it is common and natural English and (b) we know from previous studies that it engages this core LH fronto-temporal system, as opposed to other, e.g. syntactic complexity- or violation-based tasks, which tend to engage domain-general components more related to task context and off-line processing.

As to why we have chosen to use RT rather than accuracy data, in fact we collected both accuracy and RT data, but we always chose to focus on the RT data because accuracy is a blunt instrument in this kind of study. This is because there is not a correct or incorrect response to ambiguity (see long discussion on this in the previous response letter on page 2 and 4). When people hear a syntactically ambiguous sentence such as: “*When walking beside the runway, **landing planes** ARE/IS...*”, both continuations are grammatically legal but one is more strongly preferred (as we know from the statistics of the language and pre-test data, Tyler et al., 2011.). The graded nature of RTs are more sensitive to these preferences than binary accuracy judgments.

We have now added further clarification on the nature of the task and the choice of RTs to Methods (page 29, 1st paragraph).

Second, R1 raised a concern about the processing/analysis of RT data, specifically as it relates to group/age differences. Again, the authors now provide a full description of their processing approach, but their response didn't directly justify why their approach is optimal for this dataset.

Regarding our preprocessing approach of the RT data, now we added further information to the manuscript (page 30, 1st paragraph), why we chosen the specific preprocessing step described. Briefly, the method chosen, which we have used in most of our previous work (Tyler et al., 2013; Campbell et al., 2016), reduces the influence of outlying RTs without the loss of individual datapoints (Ratcliff, 1993). We opted to reduce the influence of outlying RTs rather than trim them outright as there was likely some meaning in particularly slow responses. There was variability in the dominance (or “expectedness”) of the disambiguating verb across ambiguous sentences, with some continuations (in the subordinate condition) being highly unexpected given their low frequency in the language. Overturning the dominant interpretation in these cases takes time and RTs may be particularly slow as a result. In order to retain the information contained in these trials, but at the same time reduce their influence on the mean, we used an inverse transform. Indeed, there are other methods which can achieve a similar result (e.g., Winsorizing; Erceg-Hurn & Mirosevich, 2008), but we opted for this one as we have used it many times before.

Finally, R2 raised a concern with respect the novelty of the research, and whether it is more appropriate for a specialized journal. The authors' claim regarding the novelty of the within-subject approach is certainly valid from my reading of the literature. However, I do share R2's thoughts with respect to the relevance of this work to a broader science audience.

Beyond the methodological strengths of the manuscript, reiterated in our previous response letter and also noted by the reviewer, we believe our study bears several scientific merits that make it worthy for the interest of the wider scientific community and the public.

First, our large sample size covering the entire adult lifespan allowed us to identify domain-general, domain-specific and task-negative brain components in each of the three cognitive domains investigated (see Figures 2-5). The results that 1) the identity of these different types of components is largely independent of age and that 2) they were identified both in terms of their task-specific activation/suppression and their correlation to behavioural performance, renders our study relevant not only to scientists of cognitive ageing, but also to cognitive scientists, psychologists, linguists, neuroscientists, and mental health practitioners in general.

But more generally, we believe, given the increasing demographic weight of the elderly in our ageing societies and thus the importance of understanding and promoting strategies for successful ageing, that our study addresses a topic with wide societal impact. Specifically, we believe, as part of the Cam-CAN project, that it is crucial to approach the topic of cognitive ageing with a realistic but balanced perspective, with different, domain-specific ageing trajectories across cognitive abilities, as opposed to the stereotypical view of a uniform, non-specific cognitive decline with age. We hope that our approach and results will help strengthen this more nuanced view of cognitive ageing in both the wider scientific community and the public audience.

We have now extended the discussion to make the above points more explicit (page 26, 2nd paragraph).

References:

Campbell, Karen L., et al. "Robust Resilience of the Frontotemporal Syntax System to Aging." *The Journal of Neuroscience* 36.19 (2016): 5214-5227.

Davis, Simon W., et al. "Age-related sensitivity to task-related modulation of language-processing networks." *Neuropsychologia* 63 (2014): 107-115.

Erceg-Hurn, David M., and Vikki M. Mirosevich. "Modern robust statistical methods: an easy way to maximize the accuracy and power of your research." *American Psychologist* 63.7 (2008): 591.

Griffiths, John D., et al. "Functional organization of the neural language system: dorsal and ventral pathways are critical for syntax." *Cerebral Cortex* (2012): bhr386.

Grossman, Murray, et al. "Age-related changes in working memory during sentence comprehension: an fMRI study." *Neuroimage* 15.2 (2002): 302-317.

Papoutsis, Marina, et al. "Is left fronto-temporal connectivity essential for syntax? Effective connectivity, tractography and performance in left-hemisphere damaged patients." *Neuroimage* 58.2 (2011): 656-664.

Peelle, Jonathan E., et al. "Neural processing during older adults' comprehension of spoken sentences: age differences in resource allocation and connectivity." *Cerebral Cortex* (2009): bhp142.

Ratcliff, Roger. "Methods for dealing with reaction time outliers." *Psychological bulletin* 114.3 (1993): 510.

Shafto, Meredith A., and Lorraine K. Tyler. "Language in the aging brain: the network dynamics of cognitive decline and preservation." *Science* 346.6209 (2014): 583-587.

Tyler, Lorraine K., et al. "Left inferior frontal cortex and syntax: function, structure and behaviour in patients with left hemisphere damage." *Brain* 134.2 (2011): 415-431.

Tyler, Lorraine K., et al. "Syntactic computations in the language network: characterizing dynamic network properties using representational similarity analysis." *Frontiers in psychology* 4 (2013): 271.